# Tight Bounds for Maximum Weight Matroid Independent Set and Matching in the Zero Communication Model

**Ilan Doron-Arad**
Department of Computer Science
Technion
`ilan.d.a.s.d@gmail.com`

## Abstract

Recent years have revealed an unprecedented demand for AI-based technology, leading to a common setting where immense data is distributed across multiple locations. This creates a communication bottleneck among the storage facilities, often aiming to jointly solve tasks of small solution size $k$ from input of astronomically large size $n$. Motivated by federated and distributed machine learning applications, we study two fundamental optimization problems, *maximum weight matroid independent set (MW-IS)* and *maximum weight matching (MWM)*, in a *zero communication* computational model. In this model, the data is dispersed between $m$ servers. Without any communication, each server has to send a message to a central coordinator which is required to compute an optimal solution for the original (large) instance. The goal is to minimize the size of the maximum message sent. For this natural restrictive model, we obtain deterministic algorithms that use $O(k)$-*data* per server for MW-IS and $O\left(k^2\right)$-data per server for MWM, where $k$ is the solution size (given to each server). We complement these results with tight lower bounds – ruling out any asymptotic improvement even if randomization is allowed. Our algorithms are simple and run in nearly linear time. Interestingly, we show how our zero communication algorithms yield deterministic parallel algorithms with running times $O\left(\sqrt{k} \cdot \log n\right)$ and $O\left(k^4 \cdot \log n\right)$ for MW-IS and MWM, respectively.

## 1 Introduction

Recent years have revealed an unprecedented demand for AI-based technology. Consequently, data and computing resources are often distributed across multiple locations [61, 59, 42] either due to difficulties of storing immense data in a single location [17] or privacy and security concerns [41]. Furthermore, communication among the storage facilities is often a bottleneck or prohibited.

This data storage shift has a significant effect on the relevant algorithmic model of computation. Classic sequential algorithms, where the entire data is given to a single processor, are no longer applicable to the most urgent algorithmic challenges we face today. In response, the algorithmic community focuses on the core fundamental problems on new computational models. This include distributed algorithms [44], parallel computing [54], online algorithms [2], dynamic and streaming algorithms [49], and many more computational models.

Motivated by federated and distributed machine learning applications [59, 40, 42], this work focuses on an ultra-restrictive model of computation, which allows *zero communication* between different parties (aka, servers) holding parts of large data. Each server sends without any communication a subset of its data to a central coordinator who is obligated to find an optimal solution for the original

39th Conference on Neural Information Processing Systems (NeurIPS 2025).

instance. The goal is to minimize the amount of data sent. We often consider problems where the solution size $k$ is significantly smaller than the overall input size $n$. This model is a one-round message-passing model described more formally below.

## 1.1 The Zero Communication Model

Before the definition of the model, we define the class of problems appropriate for the model. We consider the general class of weighted *subset selection problems* such as weighted matroid optimization and maximum weight matching. In a general subset selection problem $\mathcal{P}$ parameterized by solution size, an instance is described as $I = (E, w, k)$, where $E$ is a set of elements, $w : E \to \mathbb{R}$ is a weight function, and $k \in \mathbb{N}$ is a cardinality constraint (upper bound) on the solution size; we henceforth refer to $k$ as the *solution size* with a slight abuse of notation. Moreover, there exists a set $\mathcal{F} \subseteq 2^E$ of feasible solutions, which are usually not given explicitly in the input but must be inferred computationally. The goal is generally to find $S \in \mathcal{F}$ such that $|S| \leq k$ and $\sum_{e \in S} w(e)$ is optimized – either maximized or minimized. For the following definitions, fix some subset selection problem $\mathcal{P}$ and an instance $I = (E, w, k)$ of $\mathcal{P}$.

In this model, the set $E$ is partitioned $E_1, \ldots, E_m$ between $m$ servers for some $m \in \mathbb{N}$ and each server also receives the parameter $k$. In a *feasible selection* of the given instance, each $i \in \{1, \ldots, m\}$ selects $E_i' \subseteq E_i$ without any communication with any other server $j \in \{1, \ldots, m\} \setminus \{i\}$; then, server $i$ sends $\{(e, w(e))\}_{e \in E_i'}$ (that is, an explicit encoding of the selected elements and their weights) to a *central coordinator*, such that $\bigcup_{i \in \{1, \ldots, m\}} E_i'$ must contain an optimal solution for $I$: there is $S \subseteq \bigcup_{i \in \{1, \ldots, m\}} E_i'$ such that $S$ is an optimal solution for the instance. The goal is to find a feasible selection of $I$ that minimizes the maximum message *size* per server: $\max_{i \in \{1, \ldots, m\}} |E_i'|$. For some $d \in \mathbb{N}$, we refer to an algorithm in this model as a *d-data* algorithm if each server transmits at most $d$ element-weight pairs to the coordinator: $d \geq \max_{i \in \{1, \ldots, m\}} |E_i'|$.[1]

An algorithm is a *randomized* (respectively, deterministic) $d$-data algorithm if it uses $d$-data and yields a feasible selection with probability at least $\frac{1}{2}$ (probability 1 and does not use any form of randomness). The only form of randomness considered in the paper is on the feasibility (Monte Carlo) and does not consider random algorithms with a random bound on the running time (Las Vegas). We emphasize that despite the name, the zero communication model allows one-directional communication between a server and the coordinator, but prohibits communication between any two servers.

The above paragraph assumes an explicit data model, where each element and its weight are explicitly communicated. However, this model can be easily generalized such that each server $i$ sends some encoded *message* from which the coordinator can decode $\{(e, w(e))\}_{e \in E_i'}$. Due to Shannon's source coding theorem [58], it holds that $\Omega(\log |\mathcal{F}|)$ bits are required to be sent for uniquely distinguishing between all pairs of solutions in $\mathcal{F}$, which implies that for exponential size solution set $\mathcal{F}$ the explicit data encoding cannot be asymptotically improved under unique encoding. Moreover, the elements may have large metadata (e.g., large weights) and their encoding size may fluctuate dramatically. Therefore, we focus on simple explicit encoding of $\{(e, w(e))\}_{e \in E_i'}$ for the messages and use the well-defined *number* of elements sent rather than the size of the message.

As an example, consider the maximum weight matching (MWM) problem in the zero communication model. Consider a graph $G = (V, E)$ and a weight function on the edges $w : E \to \mathbb{R}$. The edge set $E$ is partitioned between $m$ servers $E_1, \ldots, E_m$. Each server $i \in \{1, \ldots, m\}$ sends a subset of the given edges and their weights $\{(e, w(e))\}_{e \in E_i'}$ to the coordinator, such that $\bigcup_{i \in \{1, \ldots, m\}} E_i'$ contains a maximum weight matching of $G$ with respect to $w$. Some of our results are tight bounds for the number of edges required to be sent from each server in this model.

**Motivation**  AI and ML applications use more than ever before data distributed across multiple locations such as data centers [60], hospitals [35], or edge devices [45]. Task assignment problems (or, matching) naturally appear in such a distributed setting: Each vertex represents an agent and the algorithm aims to select an optimal assignment of agents to tasks, where the servers hold only partial information on a subset of the agents. For example, in medical settings, patient data may be partitioned across hospitals due to privacy or logistical constraints [56, 35]. Similar scenarios occur

---

[1]An alternative objective is to minimize the total communication, resulting with a similar model.

in recommendation systems and federated learning, where we aim to solve combinatorial problems efficiently despite limited communication.

Another motivation for studying this model is its ability to derive parallel algorithms in a standard computational model, as we demonstrate in this paper. Furthermore, the zero communication model allows to distinguish between the difficulty of problems in a common distributed setting. Knowing which problems exhibit this property can have practical implications and enable robust algorithms that do not have to process large data on a single processor. Finally, the model generalizes well-studied problems in the literature, such as one-direction communication, making it a natural extension of prior work (see Section 1.3).

## 1.2 Our Results

Despite the plethora of research works on maximum weight matching and matroid independent set in various models, some fundamental questions are still present. This paper studies the elementary question of how much data from multiple servers needs to be delivered to solve a fundamental problem optimally with zero communication between the servers. This is particularly interesting when the solution size $k$ is small – the optimal solution does not require most data in the input. Our main results are tight bounds for these problems in the zero communication model and their direct implications to parallel algorithms, giving bounds that depend on the solution size $k$. We begin with formal definitions of the considered problems.

**Maximum Weight Matroid Independent Set (MW-IS)**  Consider a finite ground set $E$ and let $\mathcal{I} \subseteq 2^E$ be a non-empty set of subsets of $E$ called the *independent sets* of $E$. The set system $\mathcal{M} = (E, \mathcal{I})$ is a *matroid* if (i) for all $A \in \mathcal{I}$ and $B \subseteq A$ it holds that $B \in \mathcal{I}$; (ii) for any $A, B \in \mathcal{I}$ where $|A| > |B|$ there is $e \in A \setminus B$ such that $B \cup \{e\} \in \mathcal{I}$. In the MW-IS problem, given a matroid $(E, \mathcal{I})$ and a weight function $w : E \to I\!R$ on the elements, the goal is to find a *maximum weight independent set*: $S \in \mathcal{I}$ such that $\sum_{e \in S} w(e)$ is maximized. Additionally, this problem can be considered in a *parametric* setting, where the input also consists of a bound $k \in I\!N$ and the chosen IS has to be of cardinality bounded by $k$. We refer to this problem as $k$-MW-IS. As this variant is more general (using matroid truncation), our results are described for $k$-MW-IS and hold also for MW-IS.

**Maximum Weight Matching (MWM)**  Let $G = (V, E)$ be a graph, and let $w : E \to I\!R$ be a weight function on the edges. The goal is to find a matching $M$ (a set of pairwise disjoint edges) that maximizes $\sum_{e \in M} w(e)$. Similarly to MW-IS, a variant of this problem is $k$-MWM, where the input also consists of a bound $k \in I\!N$ on the cardinality of the maximum matching that can be selected.

Throughout the paper, $k$ denotes the size of an optimal solution for a given problem (in most cases, either MW-IS or MWM) and $n = |E|$ denotes the input size. Let $\tilde{O}, \tilde{\Omega}$ denote big-$O, \Omega$ notations, respectively, suppressing logarithmic factors in $n$. Our main algorithmic results in the zero communication model are given below. These algorithms are deterministic and run in nearly linear time. In addition, these results hold even if the inputs to some of the servers overlap. Interestingly, our algorithm for MW-IS does not require the servers to know the parameter $k$; on the other hand, this parameter is necessary for our MWM algorithm (i.e., the result holds for $k$-MWM).

**Theorem 1.1.** *There is a deterministic $k$-data algorithm for $k$-(MW-IS) and MW-IS. Moreover, for each server $i \in [m]$ the running time of server $i$ is $\tilde{O}(\sqrt{|E_i|})$ parallel time, or $\tilde{O}(|E_i|)$ non-parallel time.*

In a similar setting, we give a result for MWM as well.

**Theorem 1.2.** *There is a deterministic $O(k^2)$-data algorithm for $k$-MWM. Moreover, for each server $i \in [m]$ the running time of server $i$ is $\tilde{O}(|E_i|)$.*

We complement the above algorithms with tight lower bounds. These lower bounds rule out any qualitative improvement of our algorithms, even if randomization is allowed.

**Theorem 1.3.** *There is no randomized $o(k)$-data algorithm for $k$-MW-IS.*

We note that Theorem 1.3 holds for every $n, m, k, c \in I\!N \setminus \{0\}$ such that $n = c \cdot k \cdot m$ and $m = k$, where $n$ is the number of elements, $m$ is the number of servers, $k$ is the solution size, and $c$ is an additional integer parameter. In particular, the above result holds for arbitrarily large $n$ with

respect to $k, m$, and uses only two distinct weights. Moreover, for deterministic algorithms, the bound rules out $(k - 2)$-data algorithms and is effectively tight. Finally, the result also applies to any number of communication rounds with the coordinator and not necessarily one round of one-way communication.

Similarly to MW-IS, We give a tight lower bound for $k$-MWM.

**Theorem 1.4.** *For every $k \in \mathbb{N}$, there is no randomized $o\left(k^2\right)$-data algorithm for $k$-MWM.*

The above lower bound use weighted instances. We give an additional easy lower bound for unweighted MWM instances, or, maximum cardinality matching (MCM)) instances.

**Theorem 1.5.** *For every even $k \geq 2$ there is no deterministic $\left(\frac{k-1}{2}\right)$-data algorithm for $k$-MCM.*

The above algorithmic results in the zero communication model follow from the structural properties of the given problem. Interestingly, these algorithms lead to simple deterministic parallel algorithms for MW-IS and MWM, significantly strengthening the running time guarantee for small solution size $k$. As in various previous works on the subject (e.g., [37, 48, 50, 28]) we assume that the weights are poly-logarithmic in the size of the ground set: $w(e) = \text{polylog}(n)$ for every $e \in E$. Dropping this practical assumption would change the guarantee of our algorithms to linearly depend on the maximum representation size in bits of any of the weights. We start with MW-IS, and give a sub-linear parallel running time in the rank of the matroid (effectively equivalent to the parameter $k$ using matroid truncation).

**Theorem 1.6.** *There is a deterministic parallel $\tilde{O}(\sqrt{k})$ time algorithm for $k$-MW-IS.*

The above algorithm works for *general* matroids, unlike parallel efficient (RNC algorithms, see Section 1.3) that can be applied only to *linear matroids* [37, 48, 50].

**Theorem 1.7.** *There is a deterministic parallel $\tilde{O}\left(k^3\right)$ time algorithm for $k$–MWM.*

We note that the above statement holds for general graphs, and the running time bound may be improved for special cases of the problems, such as bipartite graphs. Using the above result as a black-box, we obtain a parallel algorithm for MWM (without a cardinality constraint).

**Theorem 1.8.** *There is a deterministic parallel $\tilde{O}\left(k^4\right)$ time algorithm for MWM.*

A summary of our main results is given in Table 1.

| Problem | Zero Communication | | Parallel Time |
|---|---|---|---|
| | Algorithm | Lower Bound | Algorithm |
| $k$-Max Weight Matroid IS | $k$ | $\Omega(k)$ | $\tilde{O}(\sqrt{k})$ |
| $k$-Max Weight Matching | $O\left(k^2\right)$ | $\Omega\left(k^2\right)$ | $\tilde{O}\left(k^3\right)$ |

Table 1: Our main results. In the table, $k$ denotes the solution size. The zero communication algorithms are deterministic and the lower bounds also rule out randomized algorithms.

## 1.3 Related Work

There has been extensive work on matching and matroid optimization in similar models to those studied in this paper. However, the majority of these works focus on approximation algorithms. In the one-round communication complexity model [29, 36], the input is partitioned between two players Alice and Bob. Alice sends a single message to Bob, and Bob outputs an answer using his input and the message received from Alice. Maximum matching in this model was first studied by Goel, Kapralov, and Khanna [29], who designed an algorithm that achieves a $\frac{2}{3}$-approximation in bipartite graphs using only $O(n)$ communication and proved that any better than $\frac{2}{3}$-approximation algorithms would require at least $n^{1+\Omega\left(\frac{1}{\log \log n}\right)}$ communication complexity. This holds even on bipartite graphs (see also [4]). Later, Lee and Singla [47] obtained a $\frac{3}{5}$-approximation algorithm for general graphs.

Assadi and Bernstein [3] obtained a 2/3-approximation for maximum matching in the one-round communication complexity; they also obtained approximation algorithms for stochastic matching

[11, 6, 5] (finding a maximum matching in a random graph) and fault-tolerance [13, 8, 12] matching (preserving an approximation guarantee even if any sufficiently small subset of edges is removed from the graph), where all of their results use the edge-degree constrained sub-graph (EDCS) matching construction originating in [10]. The EDCS matching construction has been generalized to matroid intersection by Huang et al. [32], achieving roughly the same approximation ratio of $\left(\frac{2}{3} - \varepsilon\right)$ in the one-round communication complexity model.

In the Message-Passing Model [16, 33, 53, 34], each server (or player) receives a part of the input and communication is possible between all pairs of servers. After some communication between the servers, the mutual goal is to send a message to a coordinator that computes some function of the input while minimizing the total communication complexity. For simultaneous protocols (the messages are sent simultaneously to the coordinator) Assadi et al. [7] obtained tight randomized $\alpha$-approximation for maximum matching for $\alpha \leq \frac{1}{\sqrt{m}}$ with total communication of $O(n \cdot m \cdot \alpha^2)$ and a deterministic protocol with total communication of $O(n \cdot m \cdot \alpha)$, where $n$ is the number of vertices and $m$ is the number of servers. For non-simultaneous setting, there is a tight $\Theta\left(\alpha^2 \cdot m \cdot n\right)$ information-bits $\alpha$-approximation for maximum matching [33]. We are not aware of any results for matroid optimization in this model.

As mentioned above, there are many works on bounded or one-directional communication; some of them can be seen as special cases or variants of the zero communication model. In addition, the idea of providing smaller-yet-representative dataset, known as a *coreset*, is prevalent in machine learning literature (e.g., [46, 31, 1, 22, 23, 52]). Under these lenses, our model can intuitively be viewed as a distributed (communication free) construction of non-approximate variant of a coreset. However, to the best of our knowledge, the zero communication model as defined above is introduced in this work.

In a distinct line of research, maximum matching is well-studied in parallel algorithms. The main class of parallel algorithms considered is Nick's Class (NC) – polylog$(n)$ parallel time using poly$(n)$ number of processors with unrestricted communication between them, where $n$ denotes the input size. There are randomized NC (RNC) algorithms for maximum cardinality matching and matroid intersection on linear matroids by Lovász work [43] by reducing these problems to deciding if a determinant of a symbolic matrix is equal to zero, which can be solved in NC time [15, 9]. A few years later, RNC algorithms were obtained for finding actual solutions [37, 48, 50]. There are also pseudo-deterministic NC algorithms [27, 28]; however, whether there exist deterministic NC algorithms for maximum matching (or matroid intersection) remains one of the long-standing and important questions in theoretical computer science.

## 1.4  Discussion

In this work, we provide tight bounds of $\Theta(k)$ and $\Theta\left(k^2\right)$ on the data required per server to solve $k$-MW-IS and $k$-MWM, respectively, in the zero communication model. In addition, we give deterministic $\tilde{O}\left(\sqrt{k}\right)$ and $\tilde{O}\left(k^3\right)$ time parallel algorithms for MW-IS and $k$-MWM, respectively. Some implications and limitations of our work, and some open questions are listed below.

**Matroid Intersection**  This work focuses on zero communication algorithms for MW-IS and MWM. An interesting question that can be the subject of follow-up research is what is the minimum $f(k)$ for which there is an $f(k)$-data algorithm for matroid intersection, where $k$ is the rank of one of the given matroids. An $f(k)$-data algorithm for matroid intersection can be obtained for an exponential function $f$ relatively easily. It would be intriguing to solve this for a polynomial function $f$, or show a lower bound ruling this out.

**Parallel Algorithms**  In this work, we obtain deterministic parallel algorithms for MW-IS and MWM of running time guarantee $\tilde{O}\left(\sqrt{k}\right)$ and $\tilde{O}\left(k^4\right)$, respectively. For $k \ll n$, these algorithms are faster than the classic RNC (i.e., randomized parallel) algorithms for the problems [37, 48, 50]. Moreover, our schemes handle *general* matroids rather than the special case of linear matroids as in previous works. However, for $k = \Omega\left(n^c\right)$ for some constant $c$ our parallel algorithms are no longer in NC. The most important open question here is whether a deterministic NC algorithm for MWM or MW-IS (on linear matroids) exists, even for the unweighted versions of the problems. As this question remains open for several decades, we believe that our bounds, parameterized by $k$, pave a

research path towards faster deterministic parallel algorithms for maximum matching (and matroids) with stronger parametric bounds on the solution size $k$.

**Approximation Algorithms**    All of our results apply to exact optimal versions of the considered problems. That is, unlike the majority of the previous work in similar models of low or no communication [29, 36, 16, 33, 53, 34, 7], this paper does not consider approximation algorithms. While there are results for zero communication on both matroids [32] and matching [3], most works focus on two servers (i.e., Alice sends data to Bob), or allow some communication. It would be very interesting to extend the state-of-the-art approximations for a general number $m$ of servers with zero communication. It would also be nice to obtain stronger lower bounds for such settings.

**Parameterized (fixed-parameter tractable (FPT)) Algorithms**    Our bounds are *parameterized* by $k$ – the solution size of the given problem. One may wonder, at least superficially, whether the fact that matching and matroids have such data-parameterized attributes in the zero communication model implies some connection to fixed-parameter tractable (FPT) algorithms [20]. Stated alternatively, it can be thought that the fact that $\tilde{O}(\text{poly}(k))$-data algorithms exist for MW-IS and MWM can be attributed to the fact that these problems are efficiently solvable. However, as we show in Appendix F, there are well-known problems that are (1) efficiently solvable and (2) have a small solution size $k \ll n$, yet do not have a $d$-data zero communication algorithm for $d = o(n)$.

**MWM vs. $k$-MWM**    Our zero communication algorithm focus on the cardinality constrained version of MWM, namely, $k$-MWM. We note that without this restriction, there is only an $\Omega(n)$-data algorithm for MWM (consider a single server whose sub-graph is a star graph of $\Omega(n)$ edges, which cannot determine which subset of edges to send, as any of these edges might belong to the optimal solution depending on the input to the other servers). However, the cardinality constraint has also practical justification.

For example, bipartite graphs with one side of the bipartition significantly larger than the other appear in various applications. Consider a server with $k$ processing slots and a given set of $n \gg k$ tasks, each task can be scheduled on a subset of the slot, determined by compatibility issues such as memory communication bandwidth, or other performance or security reasons. This induces a bipartite graph with $k$ vertices on one side and $n \gg k$ vertices on the other side, where the goal of the server is to compute an MWM (the weight being the price a task is willing to pay to be scheduled on a corresponding processor). Another example is allocating ads to (a few) advertisement slots. Another variant of this problem is the well-studied partial optimal transport [24, 14]. We note that our parallel MWM and $k$-MWM algorithms use a sequential $k$-MWM algorithm as a black box. A feasible research direction is to strengthen our parallel bounds using a faster black-box $k$-MWM algorithm for special cases of the problem, such as bipartite graphs. More details are given in the appendix.

**Noisy/faulty Server Inputs**    This paper studies the zero communication model, in which the input to the servers is assumed to be intact. However, this model can be generalized to tackle noisy or faulty server inputs. The noise can be either on (1) the actual element – whether it is given to the server or not, (2) the feasibility – the server may not know exactly which subsets of elements are feasible, and (3) the weights can be noisy. This of course, only makes the problem more difficult, thus our lower bounds hold; it remains open if the algorithms given in the paper can be generalized to tackle any of these scenarios. While the answer depends on the exact formulation of the noise model, our bounds are unlikely to hold as is. It remains an interesting question for follow-up research to rigorously define a variation of our model on faulty servers and give tight bounds for the discussed problems (and more) in such a model.

**Matroid Structure and Membership Oracles**    In zero communication algorithms, each server must be able to compute independence on subsets of elements it received. Without this assumption, it would be impossible to design a zero-communication algorithm that does not simply send all elements. Most matroids considered in the literature, and those with the strongest applications, are linear matroids (e.g., graphic matroids), which can be compactly defined and allow efficient independence testing with relatively small memory. However, there are also non-linear matroids, which may require exponential memory (in the number of elements of the ground set) to distinguish correctly between independent sets and non-independent sets. These matroids appear less in practical

settings and will require that each server have larger memory. To make independence testing more abstract, we assume that independence is computed via a *membership oracle* (see Section 2).

**Organization**    In Section 2 we give some preliminary definitions and notations. In Sections 3 and 4 we give our zero communication algorithms for MW-IS and MWM, respectively. Then, Section 5 describes the implications of the above algorithms for parallel computation. Due to space constraints, We give our lower bounds for zero communication algorithms and some of the remaining proofs in the appendix.

## 2    Preliminaries

**Basic Notations**    Let $I\!N = \{1, 2, \ldots\}$ be the set of natural numbers excluding zero. For any $k \in I\!N$ let $[k] = \{1, \ldots, k\}$ for short. In addition, for any set $X$, function $f : X \to I\!R$, and finite $S \subseteq X$ let $f(S) = \sum_{e \in S} f(e)$. With a slight abuse of notation, we occasionally use the same notation for a function $f$ and for a restriction of $f$ to a subset of its domain.

**Graph and Matching Notations**    Fix a graph $G = (V, E)$ for the remaining of this section. In general, all edges in this paper are undirected and with a slight abuse of notation, we use $(u, v)$ to denote the undirected edge $\{u, v\}$ between vertices $u, v \in V$. We also assume that there are no parallel edges throughout this paper. Let $N(v) = \{u \in V \mid (u, v) \in E\}$ be the set of neighbors of some $v \in V$. A *matching* in $G$ is a subset of edges $M \subseteq E$ such that for all $v \in V$ it holds that $v$ is an endpoint of at most one edge in $M$; that is, $|\{(u, v) \in M \mid u \in V\}| \leq 1$. Furthermore, $M$ is a *perfect* matching if every vertex appears exactly once as an endpoint of an edge in $M$. When the graph is clear from the context, we use $V(S) = \bigcup_{(u,v) \in S} \{u, v\}$ as the vertex set of end-points of a subset of edges $S \subseteq E$ in the graph $G$. Given $S \subseteq V$ define the induced graph $G[S] = (S, E[S])$ of $G$ and $S$, where $E[S] = \{(u, v) \in E \mid u, v \in S\}$. Finally, we use the standard notation $G = (A, B, E)$ for a bipartite graph, where $A, B$ are sets of vertices and $E \subseteq A \times B$.

**Matroids**    Let $\mathcal{M} = (E, \mathcal{I})$ be a matroid. All matroids in this paper are assumed to be general matroids whose set of elements is given in the input and for any $S \subseteq E$, determining whether $S \in \mathcal{I}$ can be done in one query to a membership oracle that is assumed to take $O(1)$ time and memory. We note that this is the standard formalism of representing general matroids for algorithmic purposes (see, e.g., [57, 51] for more details).

A *basis* of a matroid is an inclusion-wise maximal independent set of the matroid. The following matroid operations remain matroids (see, e.g., [57]).

- Matroid Restriction: For any $F \subseteq E$ define $\mathcal{I}_{\cap F} = \{A \in \mathcal{I} \mid A \subseteq F\}$ and $\mathcal{M} \cap F = (F, \mathcal{I}_{\cap F})$ as the $F$-restriction of $\mathcal{M}$.

- Contraction: For any $F \in \mathcal{I}$ define $\mathcal{I}/F = \{A \subseteq E \setminus F \mid A \cup F \in \mathcal{I}\}$ and define $\mathcal{M}/F = (E \setminus F, \mathcal{I}/F)$ as the $F$-contraction of $\mathcal{M}$.

- Truncation: For any $q \in \mathbb{N}$ define $\mathcal{I}_{\leq q} = \{A \in \mathcal{I} \mid |A| \leq q\}$ and $(E, \mathcal{I}_{\leq q})$ as the $q$-truncation of $\mathcal{M}$.

## 3    A Zero Communication Algorithm for MW-IS

In this section, we obtain a zero communication deterministic $k$-data algorithm for MW-IS. The following is a standard structural lemma used to derive our result.

**Lemma 3.1.** *Let $\mathcal{M} = (E, \mathcal{I})$ be a matroid of rank $k$, and let $w : E \to I\!R$ be a weight function. Let $m \in I\!N$, let $E_1, \ldots, E_m$ be a partition of $E$, and let $B_1, \ldots, B_m$ be maximum weight bases of the matroids $\mathcal{M} \cap E_1, \ldots, \mathcal{M} \cap E_m$, respectively. Then, there is a maximum weight basis $B^*$ of $\mathcal{M}$ such that $B^* \subseteq \bigcup_{i \in [m]} B_i$.*

We give a brief overview of the proof below. We consider a basis $B^*$ with the largest intersection with the elements in $\bigcup_{i \in [m]} B_i$. If there is some $e \in B^* \setminus \bigcup_{i \in [m]} B_i$, we can show using some insights on generalized exchange properties of matroids that there is some element $e' \in \bigcup_{i \in [m]} B_i$ with larger or equal weight w.r.t. $e$ that can be added to $B^*$ instead of $e$, implying a contradiction to the fact that

$B^*$ has a maximal intersection with $\bigcup_{i \in [m]} B_i$. This generic line of proof has also been used in other matroid and matching-related problems in approximation algorithms, including the author's prior work in a different setting [18, 19]. The full proof is given in the appendix.

We use the following result of [38] for obtaining our main result.

**Theorem 3.2.** [38] *There is a deterministic parallel $O(\sqrt{n})$ algorithm for finding a maximum weight basis of a matroid with $n$ elements.*

Thus, we obtain the following results. As our algorithm is based on simply computing a basis for each server, the parameter $k$ can be omitted from the input (that is, the following result holds for MW-IS in addition to $k$-(MW-IS)).

**Theorem 1.1.** *There is a deterministic $k$-data algorithm for $k$-(MW-IS) and MW-IS. Moreover, for each server $i \in [m]$ the running time of server $i$ is $\tilde{O}(\sqrt{|E_i|})$ parallel time, or $\tilde{O}(|E_i|)$ non-parallel time.*

*Proof.* Let $I = (E, w, k)$ be an instance of MW-IS whose elements are distributed into $m$ servers by $E_1, \ldots, E_m$. The proof follows by computing for each server $i$ a maximum weight basis $B_i$ for the matroid restricted to the set of elements $E_i$ given to server $i$ and truncated with the value $k$. These bases can be either computed using the standard greedy algorithm in time $\tilde{O}(|E_i|)$ (for more details on the greedy algorithm see, e.g., [57]), or using a parallel deterministic $\tilde{O}(\sqrt{|E_i|})$ time algorithm of Karp et al. [38]. Since the optimum is of size $k$, by Lemma 3.1 it follows that there is an optimal solution for $I$ (i.e., a maximum weight basis of the original matroid truncated by $k$) in $\bigcup_{i \in [m]} B_i$. Thus, the above algorithm optimally solves the original instance and sends at most $k$ elements from each server. The proof follows. $\qquad\square$

## 4 A Zero Communication Algorithm for Maximum Weight Matching

In this section, we give a zero communication algorithm for $k$-MWM. Our approach is based on the following auxiliary structures we call *strong sets*.

**Definition 4.1.** *Let $G = (V, E)$ be a graph, let $w : E \to \mathbb{R}$ be a weight function, and let $k \in \mathbb{N}$ be a parameter. A set of edges $S \subseteq E$ is called a strong set of $E, w$, and $k$ if for every $e = (u, v) \in E$ at least one of the following holds.*

1. *$e \in S$.*

2. *There is a matching $M \subseteq S$ of $G$ of cardinality $2 \cdot k + 1$ such that for every $f \in M$ it holds that $w(f) \geq w(e)$.*

3. *There is $x \in \{u, v\}$ and there are $2 \cdot k + 1$ distinct edges $(x, v_1), \ldots, (x, v_{2 \cdot k+1}) \in S$ such that for all $i \in [2 \cdot k + 1]$ it holds that $w((x, v_i)) \geq w((u, v))$.*

The following algorithm finds a strong set efficiently.

**Lemma 4.2.** *There is an algorithm STRONG-SET that given a graph $G = (V, E)$, a weight function $w : E \to \mathbb{R}$, and a parameter $k \in \mathbb{N}$ returns in time $O(|E| \cdot \log |E|)$ a strong set of $E, w, k$ of cardinality at most $10 \cdot k^2 + 1$.[2]*

For the entire proof of the theorem, fix a graph $G = (V, E)$, a weight function $w : E \to \mathbb{R}$, and a parameter $k \in \mathbb{N}$ and define the following algorithm STRONG-SET on the above input. Let $n = |V|$ and $m = |E|$ (with a slight abuse of notation, recall that $m$ is also used in this paper to denote the number of servers; however, this is not considered in the current algorithm and its proof). For a subset of edges $S \subseteq E$ and $v \in V$ let

$$N(S, v) = |\{(x, y) \in S \mid x = v \text{ or } y = v\}|$$

be the number of *neighbors* of $v$ in $S$. The algorithm iterates the edges in a non increasing order of weights, and adds to the constructed strong set $S$ the current edge $(u_i, v_i)$ if both $N(S, u_i), N(S, v_i)$ are bounded by $2 \cdot k$. The algorithm is formally defined as follows.

---

[2]We note that if the maximum weight $W = \max_{e \in E} |w(e)|$ is large w.r.t. $|E|$, then the running time also depends on $w$, as the running time follows from sorting an array of size $|E|$.

1. Let $e_1, \ldots, e_m$ be the edges in $E$ sorted in a non increasing order w.r.t. $w$

2. Initialize $S \leftarrow \emptyset$

3. **For** $i = 1, \ldots, m$ **do:**

    (a) Let $e_i = (u_i, v_i)$

    (b) **If** $N(S, u_i) \leq 2 \cdot k$ and $N(S, v_i) \leq 2 \cdot k$ **then:** $S \leftarrow S \cup \{e_i\}$

    (c) **If** $|S| \geq 10 \cdot k^2 + 1$ **then: return** $S$

4. **return** $S$

**Intuition**    We give below some Intuition on Algorithm STRONG-SET (a formal proof is given in the appendix). For the following, consider an execution of the algorithm on a set of edges $E$ with weight function $w$, such that the algorithm returns a subset $S$ of edges. Consider some arbitrary edge $e = (u, v) \in E$, and let us give the intuition why necessarily at least one property from (1), (2), or (3) of Definition 4.1 holds for $e$. We consider several cases here. First, clearly if $e \in S$, then property (1) trivially holds for $e$. Second, assume that (1) does not hold for e and we shall show why (2) or (3) must hold. Since e is not added to S, by Step (b) of the algorithm we can consider two sub-cases to conclude.

- $N(S, u) > 2 \cdot k$ or $N(S,v) > 2 \cdot k$. From this case, (3) easily holds since $S$ has at least $2 \cdot k + 1$ neighbors of some $x \in \{u, v\}$, where all of these neighbors are with weight at least $w(e)$ by the sorted order of the edges.

- The complementary case, where $N(S, u) \leq 2 \cdot k$ and $N(S, v) \leq 2 \cdot k$. Since $e \notin S$, then the algorithm terminated by the stopping condition of Step (c) and consequently $|S| > 10 \cdot k^2$. By Step (b) of the algorithm, for each edge $e_i = (u_i, v_i)$ it holds that $N(S, u_i) \leq 2 \cdot k + 1$ and $N(S, v_i) \leq 2 \cdot k + 1$. Therefore, the edges in $S$ induce a graph with over $10 \cdot k^2$ edges that has a maximum degree of $2 \cdot k + 1$. By a result of Han [30], in this induced graph there exists a matching of a size of over $2 \cdot k$.

The following is a structural lemma for distributed MWM using strong sets, showing the power of strong sets for zero communication algorithms.

**Lemma 4.3.** *Let $G = (V, E)$ be a graph, let $w : E \to \mathbb{R}$ be a weight function, and let $k \in \mathbb{N}$ be a parameter. Additionally, let $m \in \mathbb{N}$, and let $E_1, \ldots, E_m$ be a partition of $E$. For all $i \in [m]$ let $S_i$ be a strong set of $E_i, w, k$. Then, there is a $k$-maximum weight matching $M^*$ of $G$ such that $M^* \subseteq \bigcup_{i \in [m]} S_i$.*

**Intuition**    We give below some intuition for the above result and a formal proof is given in the appendix. In the above lemma, we are given a strong set $S_i$ for each server $i$ and need to prove that the union of $S_i$'s contains a $k$-MWM. Consider a $k$-MWM denoted by $M^*$ with the maximum size of intersection with the union of $S_i$'s. If $M^*$ is contained in this union – the central server can directly compute $M^*$. Otherwise, there is an edge $e \in M^*$ that belongs to $E_i$ for some server $i$, such that $e \notin S_i$. By Definition 4.1 and as $e \notin S_i$, one of the following holds:

1. There is a matching $M$ contained in $S_i$ of cardinality $2 \cdot k + 1$ such that every edge in $M$ has weight at least $w(e)$. Thus, as $M$ is sufficiently large and since $M$ and $M^*$ are matchings, there is $e'$ in $M$ such that $M' = M^* \cup \{e'\}$ is a matching (each endpoint of $M^*$ can appear at most once as an endpoint of $M$ and $|M| > 2 \cdot |M^*|$). Note that $w(M') \geq w(M^*)$.

2. There is an endpoint $x$ of $e$ and there are $2 \cdot k + 1$ distinct edges adjacent to $x$: $(x, v_1), ..., (x, v_{2 \cdot k+1})$, each of weight at least $w(e)$. Since $|M^*| \leq k$ it has at most $2 \cdot k$ endpoints; therefore, one of the above $2 \cdot k + 1$ edges $e' = (x, v_j)$ satisfies that $v_j$ is not an endpoint of $M^*$. Thus, since $w(e') \geq w(e)$ it holds that $M' = M^* \setminus \{e\} \cup \{e'\}$ is a matching satisfying $w(M') \geq w(M^*)$.

The above two cases yield a contradiction that $M^*$ is a $k$-MWM of maximum intersection with the union of $S_i$'s.

We can now give the main theorem of this section.

**Theorem 1.2.** *There is a deterministic $O\left(k^2\right)$-data algorithm for $k$-MWM. Moreover, for each server $i \in [m]$ the running time of server $i$ is $\tilde{O}(|E_i|)$.*

*Proof.* Let $I = (V, E, w, k)$ be an instance of MWM, where the edges are distributed into $m$ servers by $E_1, \ldots, E_m$. The proof follows by computing for each server $i$ a strong set of $E_i, w, k$ $S_i$ using Algorithm STRONG-SET. By Lemma 4.3, there is an optimal solution for $I$ (i.e., a $k$-MWM) in $\bigcup_{i \in [m]} S_i$. Thus, the above algorithm optimally solves the original instance and by Lemma 4.2 the algorithm sends at most $(10 \cdot k^2 + 1)$ edges from each server using zero communication. The running time guarantee of each server follows from Lemma 4.2. $\qquad\square$

## 5   Parallel Algorithms

In this section, we show that our zero communication schemes lead to fast parallel algorithms. The results follow from the next theorem, whose proof is given in the appendix. Recall that for our parallel algorithms that the weights are assumed to be poly-logarithmic in the input size $n$.

**Theorem 5.1.** *Let $\mathcal{P}$ be a subset selection problem that has a zero communication $f(k)$-data algorithm $\mathcal{A}$ in time $\tilde{O}(g(|E_i|))$ for each server $i \in [m]$, for some monotonic polynomial functions $f, g$, where $k$ is the solution size. Then, there is an $\tilde{O}\left(g\left(f(k)\right)\right)$ time parallel algorithm that given an instance of $\mathcal{P}$ returns a subset of the elements of size $\tilde{O}(f(k))$ that contain an optimal solution. The algorithm is deterministic if $\mathcal{A}$ is deterministic.*

We give below the pseudocode of the algorithm described in Theorem 5.1 and the full proof is given in the appendix. Let $I = (E, w, k)$ denote an instance of a subset selection problem $\mathcal{P}$, where $E$ is a set of elements, $w$ is a weight function, and $k$ is the solution size. Define the following algorithm $\mathcal{B}$ on instance $I$ based on the existence of a zero communication algorithm $\mathcal{A}$ for $\mathcal{P}$. Let $n = |E|$.

1. Let $j \leftarrow 0, E^0 \leftarrow E, n_0 \leftarrow n$
2. **While** $\frac{n_j}{4 \cdot f(k)} > 1$:

    (a) Let $m_j = \left\lceil \frac{n_j}{4 \cdot f(k)} \right\rceil$

    (b) Partition $E^j$ into servers $E_1^j, \ldots, E_m^j$ where $\left| E_i^j \right| \leq 4 \cdot f(k)$ for all $i \in [m_j]$

    (c) Execute $\mathcal{A}$ on the instance $I_j = \left\{ (E_i^j, w, k) \right\}_{i \in [m_j]}$

    (d) Let $K_i^j \subseteq E_i^j$ be the elements brought from server $i$ to the central coordinator $\forall i \in [m_j]$

    (e) Update $E^{j+1} \leftarrow \bigcup_{i \in [m_j]} K_i^j, n_{j+1} \leftarrow \left| E^{j+1} \right|, j \leftarrow j + 1$

3. Return $E^j$

Intuitively, at each iteration of the algorithm, each server applies a filtering of the elements it receives: Each server gets roughly $4 \cdot f(k)$ elements and returns only $f(k)$ elements (in parallel to the other servers). Thus, in $O(\log n)$ such rounds, we obtain an $O(f(k))$ size instance. We note that the number of servers $m_j$ used in each iteration decreases exponentially.

By Theorem 5.1, we have the following results for MW-IS and MWM.

**Theorem 1.6.** *There is a deterministic parallel $\tilde{O}(\sqrt{k})$ time algorithm for $k$-MW-IS.*

*Proof.* By Theorem 1.1, there is a deterministic zero communication $k$-data algorithm for maximum weight independent set of a matroid in deterministic parallel time of $\tilde{O}(\sqrt{|E_i|})$ for each server $i$. Assuming a membership oracle for the given matroid, sending each element to the central coordinator takes $\tilde{O}(1)$ bits as each element-weight pair in the ground set can be encoded in $O(\log n)$ bits assuming pseudo-polynomial weights. Thus, sending $k$ elements from each server can be done using $\tilde{O}(k)$ bits. Hence, by Theorem 5.1 there is a deterministic parallel $\tilde{O}(\sqrt{k})$ algorithm that computes a subset of elements of size $\tilde{O}(k)$ containing an optimal solution; on this subset, we can apply Lemma 3.2 to get an overall $\tilde{O}(\sqrt{k})$ parallel time deterministic algorithm for MW-IS. $\qquad\square$

The proofs of Theorem 1.7 and Theorem 1.8 are given in the appendix.

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

# A  Zero communication Algorithm for MW-IS

In this section, we give the proof of Lemma 3.1.

**Lemma 3.1.** *Let $\mathcal{M} = (E, \mathcal{I})$ be a matroid of rank $k$, and let $w : E \to \mathbb{R}$ be a weight function. Let $m \in \mathbb{N}$, let $E_1, \ldots, E_m$ be a partition of $E$, and let $B_1, \ldots, B_m$ be maximum weight bases of the matroids $\mathcal{M} \cap E_1, \ldots, \mathcal{M} \cap E_m$, respectively. Then, there is a maximum weight basis $B^*$ of $\mathcal{M}$ such that $B^* \subseteq \bigcup_{i \in [m]} B_i$.*

*Proof.* Let $\mathcal{B}$ be the set of all maximum weight bases of $\mathcal{M}$. Note that $\mathcal{B} \neq \emptyset$ since every matroid has at least one basis. For every $B \in \mathcal{B}$, define $p(B) = \left| B \cap \left( \bigcup_{i \in [m]} B_i \right) \right|$ to be the *proximity* of $B$. Let

$$B^* = \arg\max \left\{ p(B) \mid B \in \mathcal{B} \right\}$$

be a maximum weight basis of $\mathcal{M}$ with maximum proximity and choose $B^*$ arbitrarily if there is more than one maximum weight basis of maximum proximity. Since $\mathcal{B} \neq \emptyset$, it holds that $B^*$ is well-defined. Assume towards a contradiction that $p(B^*) < |B^*|$; thus, there is $e \in B^* \setminus \bigcup_{i \in [m]} B_i$. Let $i \in [m]$ be the unique index such that $e \in E_i$ and let $X = \{e' \in B_i \mid w(e') \geq w(e)\}$. We use the following *strong basis exchange* property of general matroids; for more details see, e.g., [57, 51].

**Claim A.1.** *Let $A, B$ be two bases of a matroid $\mathcal{T} = (F, \mathcal{J})$. Then, for every $a \in A \setminus B$ there is $b \in B \setminus A$ such that $(A \setminus \{a\}) \cup \{b\} \in \mathcal{J}$ and $(B \setminus \{b\}) \cup \{a\} \in \mathcal{J}$.*

The next claim follows from repeated application of the exchange property of matroids. We give a proof for completeness.

**Claim A.2.** *Let $B$ be a basis and let $A$ be an independent set of a matroid $\mathcal{T} = (F, \mathcal{J})$. Then, there is $S \subseteq B \setminus A$ such that $A \cup S$ is a basis of $\mathcal{T}$.*

*Proof.* We prove the claim by induction on $t = |B| - |A|$. For the base case, $t = 0$, that is, $A$ is a basis of $\mathcal{T}$ and the proof follows for $S = \emptyset$. Assume that for all independent sets $A' \in \mathcal{J}$ with $|A'| > |A|$ there is $S' \subseteq B \setminus A'$ such that $A' \cup S'$ is a basis of $\mathcal{T}$. Now, for the step of the induction assume that $|B| - |A| > 0$ thus $|B| > |A|$. Since $A, B \in \mathcal{J}$, by the exchange property of $\mathcal{T}$ there is $e \in B \setminus A$ such that $A \cup \{e\} \in \mathcal{J}$. Let $A' = A \cup \{e\}$; as $|A'| > |A|$, by the assumption of the induction there is $S' \subseteq B \setminus A'$ such that $A' \cup S'$ is a basis of $\mathcal{T}$. Let $S = S' \cup \{e\}$. Note that $S \subseteq B \setminus A$ and that $A \cup S$ is a basis of $\mathcal{T}$, which gives the statement of the claim. ⌟

A *cycle* of a matroid $\mathcal{T} = (F, \mathcal{J})$ is $C \subseteq F$ such that $C \notin \mathcal{J}$ and for all $C' \subset C$ it holds that $C' \in \mathcal{J}$. We use the following fundamental result on cycles in matroids. The reader is referred to, e.g., [57, 51] for more details.

**Claim A.3.** *Let $\mathcal{T} = (F, \mathcal{J})$ be a matroid, let $A \in \mathcal{J}$, and let $f \in F \setminus A$ such that $A \cup \{f\} \notin \mathcal{J}$. Then, there is exactly one cycle $C \subseteq A \cup \{f\}$ of $\mathcal{T}$.*

The following is a core claim in the analysis.

**Claim A.4.** $X \cup \{e\} \notin \mathcal{I}$.

*Proof.* Assume towards a contradiction that $X \cup \{e\} \in \mathcal{I}$ and consider two cases. First, if $B_i \cup \{e\} \in \mathcal{I}$. Since $e \notin B_i$ and $B_i \cup \{e\} \subseteq E_i$, this is immediately a contradiction since $B_i$ is a basis of $\mathcal{M} \cap E_i$ and cannot be extended. Second, assume that $B_i \cup \{e\} \notin \mathcal{I}$; by the assumption, $X \cup \{e\} \in \mathcal{I}$. By Claim A.3 there is exactly one cycle $C \subseteq B_i \cup \{e\}$ and since $X \cup \{e\} \in \mathcal{I}$ we conclude that there exists some element $e'' \in B_i \setminus (X \cup \{e\})$ that belong to the unique cycle $C$. Let $B_i' = (B_i \cup \{e\}) \setminus \{e''\}$. Then, since $e''$ belongs to the unique cycle $C$, it follows that $B_i' \in \mathcal{I}$. Since $e'' \in B_i \setminus X$ it holds that $w(e) > w(e'')$ implying that

$$w(B_i') = w(B^*) + w(e) - w(e'') > w(B_i).$$

Since $|B_i'| = |B_i|$ (recall that $e \in E_i \setminus B_i$) it follows that $B_i'$ is a basis of $\mathcal{M} \cap E_i$ (any independent set of cardinality equals to the cardinality of some basis is also a basis [57]) with weight strictly larger than the weight of $B_i$. This is a contradiction since $B_i$ is a maximum weight basis of $\mathcal{M} \cap E_i$. We conclude that $X \cup \{e\} \notin \mathcal{I}$. ⌟

Using the above claims, we can complete the proof of the lemma. Since $B_i$ is a basis of $\mathcal{M} \cap E_i$, it also follows that $B_i \in \mathcal{I}$; thus, as $X \subseteq B_i$, using the hereditary property it follows that $X \in \mathcal{I}$. Then, as $B^*$ is a basis of $\mathcal{M}$, by Claim A.2 there is $S \subseteq B^* \setminus X$ such that $X \cup S$ is a basis of $\mathcal{M}$.

**Claim A.5.** $e \in B^* \setminus (X \cup S)$.

*Proof.* Recall that $e \notin \bigcup_{i \in [m]} B_i$ and in particular $e \notin X$. In addition, assume towards a contradiction that $e \in S$; thus, $X \cup \{e\} \subseteq X \cup S$. Since $X \cup S$ is a basis of $\mathcal{M}$ it holds that $X \cup S \in \mathcal{I}$; using the hereditary property it follows that $X \cup \{e\} \in \mathcal{I}$ in contradiction to Claim A.4. Overall, we conclude that $e \notin X \cup S$ and the claim follows. ⌟

By Claim A.5 it holds that $e \in B^* \setminus (X \cup S)$. Therefore, by Claim A.1 there is $e' \in (X \cup S) \setminus B^*$ such that $(B^* \cup \{e'\}) \setminus \{e\} \in \mathcal{I}$. Let $\tilde{B} = (B^* \cup \{e'\}) \setminus \{e\}$. Since $S \subseteq B^*$ and $e' \in (X \cup S) \setminus B^*$ it follows that $e' \in X$. Therefore, $w(e') \geq w(e)$ implying that

$$w\left(\tilde{B}\right) = w(B^*) + w(e') - w(e) \geq w(B^*).$$

Since $\left|\tilde{B}\right| = |B^*|$ we conclude that $\tilde{B}$ is also a maximum weight basis of $\mathcal{M}$, i.e., $\tilde{B} \in \mathcal{B}$. Since $e \notin \bigcup_{i' \in [m]} B_{i'}$ and $e' \in X$, where $X \subseteq B_i$, it follows that $p\left(\tilde{B}\right) > p(B^*)$. This is a contradiction to the maximality of $B^*$ with respect to $p$ and the proof follows. □

# B  Zero communication Algorithm for MWM

In this section, we give the remaining proofs from Section 4. We start with the analysis of the STRONG-SET algorithm.

## B.1  Analysis of Algorithm Strong-Set

**Lemma 4.2.** *There is an algorithm* STRONG-SET *that given a graph $G = (V, E)$, a weight function $w : E \to \mathbb{R}$, and a parameter $k \in \mathbb{N}$ returns in time $O\left(|E| \cdot \log |E|\right)$ a strong set of $E, w, k$ of cardinality at most $10 \cdot k^2 + 1$.*[3]

*Proof.* We give the running time and correctness analysis of Algorithm STRONG-SET below.

**Claim B.1.** *The running time of* STRONG-SET *is $O(m \cdot \log m)$.*

*Proof.* Sorting the edges takes $O(m \cdot \log m)$ time. In addition, the algorithm makes one pass over the sorted edges and in each iteration $i \in [m]$ calculates $N(S, u_i), N(S, v_i)$ and performs $O(1)$ additional operations. Clearly, we can compute $N(S, v)$ for every $v \in V$ in time $O(1)$ per iteration. We initialize $N(S, v) = 0$ for every $v \in V$; Let $i \in [m]$. If the edge $e_i = (u_i, v_i)$ is added to $S$, we update $N(S, u_i) \leftarrow N(S, u_i) + 1$, $N(S, v_i) \leftarrow N(S, v_i) + 1$, and otherwise, we do not need to update any neighbor set. Thus, each iteration takes $O(1)$ time and overall the running time of the algorithm is $O(m \cdot \log m) + O(m) = O(m \cdot \log m)$. ⌟

We prove the following invariant of the algorithm.

**Claim B.2.** *For every $v \in V$ it holds that $N(S, v) \leq 2 \cdot k + 1$ at any iteration.*

*Proof.* The algorithm starts with $S = \emptyset$ and consequently $N(S, v) = 0$ for every $v \in V$ in the first iteration. For some $i \in [m]$, assume that $N(S, v) \leq 2 \cdot k + 1$ for every $v \in V$ before iteration $i$ of the algorithm; we show that the above property still holds after iteration $i$. Consider the following cases. Note that if the algorithm terminated before iteration $i$ there is nothing to prove. Thus, assume that the algorithm did reach iteration $i$. First, assume that $N(S, u_i) \leq 2 \cdot k$ and $N(S, v_i) \leq 2 \cdot k$; therefore, after iteration $i$ it holds that $N(S, u_i) \leq 2 \cdot k + 1$ $N(S, v_i) \leq 2 \cdot k + 1$ and for every $x \in V \setminus \{u_i, v_i\}$

---

[3]We note that if the maximum weight $W = \max_{e \in E} |w(e)|$ is large w.r.t. $|E|$, then the running time also depends on $w$, as the running time follows from sorting an array of size $|E|$.

it holds that $N(S, x)$ does not change in iteration $i$. Second, assume that $N(S, u_i) > 2 \cdot k$ or $N(S, v_i) > 2 \cdot k$. Thus, in this case, $e_i$ is not added to $S$, and $S$ remains the same before and after iteration $i$. It follows that $N(S, v) \leq 2 \cdot k + 1$ for every $v \in V$ in both cases. ⌙

We also use the following result of Han [30].

**Claim B.3.** *In every graph with $M$ edges and maximum degree $\Delta$ there is a matching of size $\frac{4 \cdot M}{5 \cdot \Delta + 3}$.*

We can now complete the proof, by showing that $S$, the set returned by the algorithm, is a strong set. By Definition 4.1, let $e' = (u, v) \in E$ and consider the following cases. First, if $e' \in S$, then $e'$ satisfies the first property of Definition 4.1 and there is nothing to prove. Then, for the following assume that $e' \notin S$. Therefore, one of the following holds.

Let $i \in [m]$ such that $e_i = e'$ (there is exactly one such iteration). Since $e_i \notin S$, then in iteration $i$ of the algorithm one of the following holds.

- The algorithm returns $S$ before iteration $i$. Then, $|S| \geq 10 \cdot k^2 + 1$. In addition, by Claim B.2 the maximum degree in the induced graph $G[V(S)]$ is at most $2 \cdot k + 1$. Thus, by Claim B.3 there is a matching in $S$ of cardinality at least

$$\frac{4 \cdot |S|}{5 \cdot (2 \cdot k + 1) + 3} \geq \frac{4 \cdot (10 \cdot k^2 + 1)}{10 \cdot k + 8} = \frac{40 \cdot k^2 + 4}{10 \cdot k + 8} \geq \frac{40 \cdot k^2}{18 \cdot k} = \frac{40 \cdot k}{18} > 2 \cdot k.$$

  Since the size of a maximum matching in $G[V(S)]$ is always an integer, the above implies that there is a matching $M$ in $G[V(S)]$ of cardinality at least $2 \cdot k + 1$. Note that $w(e) \geq w(e_i)$ for every $e \in S$ due to the non increasing order of weights of edges inserted to $S$. Therefore, there is a matching $M \subseteq S$ such that $w(e) \geq w(e_i)$ for every $e \in M$, which satisfies the second property of Definition 4.1.

- The algorithm reaches iteration $i$. Then, since $e_i \notin S$, we conclude that $N(S, u_i) \geq 2 \cdot k + 1$ or $N(S, v_i) \geq 2 \cdot k + 1$ during iteration $i$. Therefore, there is $x \in \{u_i, v_i\}$ such that $N(S, x) \geq 2 \cdot k + 1$. Note that $w(e) \geq w(e_i)$ for every $e \in S$ due to the non increasing order of weights of the edges inserted to $S$. Thus, there are $2 \cdot k + 1$ distinct edges $(x, y_1), \ldots, (x, y_{2 \cdot k + 1}) \in S$ such that $w((x, y)) \geq w(e_i)$ for every $y \in \{y_1, \ldots, y_{2 \cdot k + 1}\}$. This satisfies the third property of Definition 4.1.

To conclude the proof, we show a second trivial invariant on the size of $S$.

**Claim B.4.** $|S| \leq 10 \cdot k^2 + 1$ *at the end of the algorithm.*

*Proof.* The algorithm starts where $S = \emptyset$ and consequently $|S| = 0 < 10 \cdot k^2 + 1$. For some $i \in [m]$, assume that before iteration $i$ it holds that $|S| < 10 \cdot k^2 + 1$; we show that either (i) the above property either holds after iteration $i$ or (ii) the algorithm terminates and $|S| = 10 \cdot k^2 + 1$. Consider the following cases. Note that, if the algorithm terminated before iteration $i$ assuming $|S| < 10 \cdot k^2 + 1$ before the iteration, there is nothing to prove. Thus, assume that the algorithm did not terminate before iteration $i$. In iteration $i$, consider the following cases. First, $e_i \notin S$; then, the size of $S$ did not change and the claim follows. Second $e_i \in S$; then, $|S|$ increased by 1; therefore, either (i) $S$ still satisfies that $|S| < 10 \cdot k^2 + 1$, or (ii), $|S| = 10 \cdot k^2 + 1$ which terminates the algorithm. In both cases, the proof follows. ⌙

The above implies the proof of the lemma. □

## B.2  Structural Lemma for matching

**Lemma 4.3.** *Let $G = (V, E)$ be a graph, let $w : E \to \mathbb{R}$ be a weight function, and let $k \in \mathbb{N}$ be a parameter. Additionally, let $m \in \mathbb{N}$, and let $E_1, \ldots, E_m$ be a partition of $E$. For all $i \in [m]$ let $S_i$ be a strong set of $E_i, w, k$. Then, there is a $k$-maximum weight matching $M^*$ of $G$ such that $M^* \subseteq \bigcup_{i \in [m]} S_i$.*

*Proof.* Let $\mathcal{M}$ be the set of all $k$-MWMs of $G$ and $w$. Note that $\mathcal{M} \neq \emptyset$ since every graph has at least one matching: the empty set is in particular a $k$-matching with weight 0. For every $M \in \mathcal{M}$ define $p(M) = \left| M \cap \left( \bigcup_{i \in [m]} S_i \right) \right|$ to be the *proximity* of $M$. Let

$$M^* = \arg\max \left\{ p(B) \mid B \in \mathcal{M} \right\}$$

be a $k$-MWM with maximum proximity. Assume towards a contradiction that $p(M^*) < |M^*|$; thus, there is $e \in M^* \setminus \bigcup_{i \in [m]} S_i$. Let $i \in [m]$ be the unique index such that $e \in E_i$ and let $e = (u, v)$ for $u, v \in V$. We use the following claims.

**Claim B.5.** *For every matchings $A, B$ of $G$ such that $|A| > 2 \cdot |B|$ there is $a \in A \setminus B$ such that $B \cup \{a\}$ is a matching of $G$.*

*Proof.* Since $B$ is a matching of $G$ it holds that $|V(B)| \leq 2 \cdot |B| < |A|$. Thus, since each $v \in V(A)$ appears at most once as an endpoint of an edge in $A$, there is $(x, y) \in A$ such that $x, y \notin V(B)$. Thus, $B \cup \{(x, y)\}$ is a matching of $G$. ⌟

The next claim leads directly to the proof of the lemma.

**Claim B.6.** *There is $e^* \in S_i$ such that $(M^* \setminus \{e\}) \cup \{e^*\}$ is a matching of $G$ and $w(e^*) \geq w(e)$.*

*Proof.* Since $S_i$ is a strong set of $E_i, w, k$ and $e \in E_i \setminus S_i$, one of the following holds by Definition 4.1.

1. There is a matching $M \subseteq S_i$ of $G$ of cardinality $2 \cdot k + 1$ such that for all $f \in M$ it holds that $w(f) \geq w(e)$. Observe that $|M| \geq 2 \cdot k + 1$ and $|M^*| \leq k$; thus, $|M| > 2 \cdot |M^*|$ and note that both $M, M^*$ are matchings. Then, by Claim B.5 there is $e^* \in M$ such that $(M^* \setminus \{e\}) \cup \{e^*\}$ is a matching of $G$; by the definition of $M$, it holds that $w(e^*) \geq w(e)$. Hence, $e^*$ implies the statement of the claim in this case.

2. There is $x \in \{u, v\}$ and there are $2 \cdot k + 1$ distinct edges $(x, v_1), \ldots, (x, v_{2 \cdot k + 1}) \in S_i$ such that for all $i \in [2 \cdot k + 1]$ it holds that $w((x, v_i)) \geq w((u, v))$. Since $|M^*| \leq k$ it holds that $|V(M^*)| \leq 2 \cdot k$; therefore, as each $v' \in \{v_1, \ldots, v_{2 \cdot k + 1}\}$ can appear as an endpoint of an edge in $M^*$ at most once, there is $j \in [2 \cdot k + 1]$ such that $v_j \notin V(M^*)$. Thus, $(M^* \setminus \{e\}) \cup \{(x, v_j)\}$ is a matching of $G$ such that $w((x, v_j)) \geq w(e)$. Hence, the proof follows with $e^* = (x, v_j)$.

The above two cases give the statement of the claim. ⌟

By Claim B.6, let $\tilde{M} = (M^* \setminus \{e\}) \cup \{e^*\}$, where $e^* \in S_i$ such that there is $e^* \in S_i$ satisfying that $\tilde{M}$ is a matching of $G$ and $w(e^*) \geq w(e)$. Thus, $\tilde{M}$ is a matching of $G$ such that

$$w\left( \tilde{M} \right) = w(M^*) + w(e^*) - w(e) \geq w(M^*).$$

Hence, $M^* \in \mathcal{M}$. Since $e^* \in \bigcup_{i \in [m]} S_i$ and $e \notin \bigcup_{i \in [m]} S_i$ it follows that $p\left( \tilde{M} \right) > p(M^*)$ in contradiction to the maximality of $p(M^*)$. □

## C  Lower Bound for MW-IS

In the following sections, we give the proofs of our lower bounds. These bounds are information-theoretic bounds on the number of elements required to be sent from each server in order to solve the given problem optimally.

**Theorem 1.3.** *There is no randomized $o(k)$-data algorithm for $k$-MW-IS.*

*Proof.* Let any $n, m, k, c \in \mathbb{N} \setminus \{0\}$ such that $n = c \cdot k \cdot m$ and $m = k$. As before, $n$ will be the number of elements, $m$ the number of servers, $k$ the solution size, and $c$ is an additional parameter. For any $i \in [m]$, create the $i$-th *server*, a set of elements $E_i = \{e_1^i[\ell], \ldots, e_k^i[\ell] \mid \ell \in [c]\}$ and let $E = \bigcup_{i \in [m]} E_i$; note that $|E| = n$. Moreover, define $E^j = \{e_j^i[\ell] \mid i \in [m], \ell \in [c]\}$ for all $i \in [m]$

and $j \in [k]$. For every $i, j \in [k]$ and $\ell \in [c]$ define the weight of $e_j^i[\ell]$ as $w\left(e_j^i[\ell]\right) = 1 + \frac{j}{2 \cdot k^2}$; let $W_j = w\left(e_j^i[\ell]\right)$ for simplicity. We use the next trivial claim.

**Claim C.1.** *For all $j_1, j_2 \in [k]$ such that $j_1 < j_2$ the following holds.*

- $k \cdot W_{j_1} < k \cdot W_{j_2}$.

- $k \cdot W_{j_1} > (k-1) \cdot W_{j_2}$.

*Proof.* Clearly, $W_{j_1} = 1 + \frac{j_1}{2 \cdot k^2} < 1 + \frac{j_2}{2 \cdot k^2} = W_{j_2}$; thus, $k \cdot W_{j_1} < k \cdot W_{j_2}$. In addition,

$$k \cdot W_{j_1} = k + k \cdot \frac{j_1}{2 \cdot k^2} \geq k = (k-1) + 1 > (k-1) + \frac{k}{2 \cdot k} \geq (k-1) + \frac{j_2}{2 \cdot k}$$

$$= (k-1) + (k-1) \cdot \frac{j_2}{2 \cdot k \cdot (k-1)} \geq (k-1) \cdot W_{j_2}.$$

The third inequality holds because $j_2 \in [k]$. ⌟

Note that the above defines a family of MW-IS instances with a ground set $E$ and weight function $w : E \to \mathbb{R}$ that differ by distinct selections of the set of independent sets $\mathcal{I} \subseteq 2^E$ (assuming fixed $n, m$, and $k$). For any $X \subseteq [k]$ define

$$\mathcal{I}_X = \left\{ S \subseteq E \;\middle|\; \left|S \cap E^j\right| \leq k \; \forall j \in X \text{ and } \left|S \cap E^j\right| \leq k - 1 \; \forall j \in [k] \setminus X \right\}. \tag{1}$$

Since $E^1, \ldots, E^k$ is a partition of $E$, it follows that $(E, \mathcal{I}_X)$ is a partition matroid, which is well known to be a matroid (see, e.g., [57, 51]). For any $X \subseteq [k]$, let $\Phi_X$ be the MW-IS instance defined by the matroid $(E, \mathcal{I}_X)$, partition $E_1, \ldots, E_m$ into servers, and the weight function $w : E \to \mathbb{R}$.

We prove the theorem below. For the sake of brevity, we provide a unified proof of the lower bound for both deterministic and randomized algorithms. The necessary small modifications in the proof required for randomized algorithms will appear in parentheses.

Assume towards a contradiction that there is a (randomized) $d$-data algorithm $\mathcal{A}$ for MW-IS for $d \leq k - 2$ (for $d < \frac{k-2}{2}$). For every $j \in [k]$, let $\mathcal{S}_j = \left\{ S \subseteq E^j \;\middle|\; |S| = k \right\}$. By (1) and Claim C.1, for every $j \in [k]$ it holds that $\mathcal{S}_j$ is the set of all maximum weight (with respect to $w$) bases of $(E, \mathcal{I}_X)$ for all non-empty sets $X \subseteq [k]$ such that $j = \arg\max_{j' \in X} j'$; that is, $\mathcal{S}_j$ is the set of optimal solutions of all such instances $\Phi_X$.

Consider the MW-IS instance $\Phi_{\{1\}}$. Let $\mathcal{Q} \subseteq 2^E$ be all sets queried (with probability at least $\frac{1}{2}$) by the membership oracle of the central coordinator during the execution of Algorithm $\mathcal{A}$ on instance $\Phi_{\{1\}}$, and let $E' \subseteq E$ be the (random) collection of elements brought to the central coordinator in this execution. Let

$$\mathcal{B} = \{j \in [k] \mid \mathcal{Q} \cap \mathcal{S}_j \neq \emptyset\}$$

be all indices on which Algorithm $\mathcal{A}$ queried on instance $\Phi_{\{1\}}$ at least one set in $\mathcal{S}_j$ (with probability at least $\frac{1}{2}$). Since $\mathcal{A}$ is a $d$-data algorithm for $d \leq k - 2$ (for $d < \frac{k-2}{2}$), it follows that $|E'| \leq m \cdot d$ (for all realizations). Recall that $E^1, \ldots, E^k$ are disjoint; thus, there can be at most $d$ indices $j \in [k]$ such that $\mathcal{Q} \cap \mathcal{S}_j \neq \emptyset$ (with probability at least $\frac{1}{2}$). In other words, $|\mathcal{B}| \leq d \leq k - 2$ (for randomized algorithms: $|\mathcal{B}| \leq 2 \cdot d \leq k - 2$ since for each $j \in \mathcal{B}$ the expected number of elements taken to the coordinator from $E^j$ is at least $\frac{k}{2}$ and the overall number of elements is bounded by $m \cdot d$ on all realizations). Thus, there is $j^* \in [k] \setminus 1$ such that $j^* \notin \mathcal{B}$. Observe that the output of the central coordinator in the execution of Algorithm $\mathcal{A}$ on $\Phi_{\{1\}}$ is determined by the set of elements $E'$ and the results of the queries (and the random bits of the algorithm). By (1) and Claim C.1, it follows that $\mathcal{A}$ returns a set in $\mathcal{S}_1$ (with probability at least $\frac{1}{2}$). On the other hand, for the instance $\Phi_{\{1,j^*\}}$ the set of elements in the central coordinator is also $E'$ and the result of each query except for subsets in $\mathcal{S}_{j^*}$ is the same for the membership oracles of $\mathcal{I}_{\{1\}}$ and $\mathcal{I}_{\{1,j^*\}}$. Therefore, since $\mathcal{A}$ does not query sets in $\mathcal{S}_{j^*}$ also on instance $\Phi_{\{1,j^*\}}$ (with probability at least $\frac{1}{2}$), Algorithm $\mathcal{A}$ has to return on this instance the same output it returns on instance $\Phi_{\{1\}}$ (with probability at least $\frac{1}{2}$), which is in $\mathcal{S}_1$. Since $\mathcal{S}_1 \cap \mathcal{S}_{j^*} = \emptyset$, it follows that $\mathcal{A}$ does not return an optimal solution for instance $\Phi_{\{1,j^*\}}$ (with probability at least $\frac{1}{2}$), in contradiction to the definition of $\mathcal{A}$. □

# D  Lower Bounds for MWM

We give below our two lower bounds for MWM. We start with the lower bound for unweighted instances.

## D.1  Lower Bound for Unweighted Instances

We remark that with a factor of 2 the following result can be adapted for randomized algorithms.

**Theorem 1.5.** *For every even $k \geq 2$ there is no deterministic $\left(\frac{k-1}{2}\right)$-data algorithm for $k$-MCM.*

*Proof.* Let $k \in \mathbb{N}$ such that $k \geq 2$ and let $V = [k] \times \{1\}$ and $U = [k] \times \{2\}$ be disjoint sets of vertices. Let $\sigma : [k] \to [k]$ be a bijection. Define the bipartite graph $G_\sigma = (V \cup U, E_\sigma)$ where

$$E_\sigma = \left\{ ((v,1),(u,2)) \in V \times U \mid v \leq \sigma(u) \right\}.$$

Let $P_\sigma^1, \ldots, P_\sigma^m$ be a partition of $E_\sigma$ into servers such that for all $v \in [m]$ it holds that

$$P_\sigma^v = \{((x,1),(y,2)) \in E_\sigma \mid x = v\}.$$

For every bijection $\sigma : [k] \to [k]$ define

$$M_\sigma = \left\{ ((v,1),(u,2)) \in V \times U \mid v = \sigma(u) \right\}. \tag{2}$$

We show that $M_\sigma$ is a maximum (cardinality) matching in $G_\sigma$.

**Claim D.1.** *For every bijection $\sigma : [k] \to [k]$ it holds that $M_\sigma$ is a matching in $G_\sigma$ of cardinality $k$.*

*Proof.* We first show that $M_\sigma \subseteq E_\sigma$. For every $((v,1),(u,2)) \in V \times U$ such that $v = \sigma(u)$, in particular it holds that $v \leq \sigma(u)$. This implies that $((v,1),(u,2)) \in E_\sigma$ which consequently means that $M_\sigma \subseteq E_\sigma$. It remains to show that $M_\sigma$ is a matching in $G_\sigma$ of cardinality $k$. Since $\sigma$ is a bijection, for each $v \in [k]$ there is exactly one $u \in [k]$ such that $v = \sigma(u)$, and for each $u \in [k]$ there is exactly one $v \in [k]$ such that $v = \sigma(u)$. Therefore, for each $v \in [k]$ and $u$ such that $v = \sigma(u)$, the vertices $(v,1),(u,2)$ appear exactly once as endpoints in $M_\sigma$. Thus, $M_\sigma$ is a matching in $G_\sigma$ of cardinality $k$  ⌟

**Claim D.2.** *Let $\sigma : [k] \to [k]$ be a bijection and let $F$ be a matching in $G_\sigma$ such that $|F| \geq k$. Then, it holds that $F = M_\sigma$.*

*Proof.* Consider two cases. First, $|F| > k$. Then, by the pigeonhole principle there is $v \in V$ that appears as an endpoint of at least two edges in $F$, implying that $F$ is not a matching. Contradiction Second, $F = k$ and every $(v,1) \in V$ appears as exactly one endpoint of an edge in $F$. Since $\sigma$ is a bijection, the inverse function $\sigma^{-1} : [k] \to [k]$ is well-defined. We prove that for each $v \in [k]$, by backward induction on $v$, that it holds that $((v,1),(\sigma^{-1}(v),2)) \in F$; since $|F| = k$, it directly follows that $F = M_\sigma$. For the base case, assume that $v = k$. Then, as $((v,1),(u,2)) \in F$ for some $u \in [k]$ (recall that $|F| = k$), by the definition of $E_\sigma$ it must hold that $v \leq \sigma(u)$; moreover, $\sigma(u) \in [k]$ since the range of $\sigma$ is also $[k]$, which implies that $v \geq \sigma(u)$. Hence, $v = \sigma(u)$ implying $u = \sigma^{-1}(v)$, and the base case follows. For the step of the induction, assume, for some $v \in [k]$, that for every $v' \in [k] \setminus [v]$ it holds that $((v,1),(\sigma^{-1}(v),2)) \in F$. Let $((v,1),(u,2)) \in F$ for some $u \in [k]$ (again, recall that $|F| = k$ hence there is such $u$). By the definition of $E_\sigma$ it holds that $v \leq \sigma(u)$. Also, by the assumption of the induction, for every $v' \in [k]$ such that $v' > v$ it holds that $((v,1),(\sigma^{-1}(v),2)) \in F$; thus, $\sigma(u) \in [k] \setminus \{v' \in [k] \mid v' > v\} = [v]$. Therefore, $v \geq \sigma(u)$ implying $v = \sigma(u)$ as required. We conclude that $F = M_\sigma$.  ⌟

Assume towards a contradiction that there is a $\left(\frac{k-1}{2}\right)$-data algorithm $\mathcal{A}$ for unweighted MWM with zero communication. For every bijection $\sigma : [k] \to [k]$ and $v \in [m]$, let $E_\sigma^v \subseteq P_\sigma^v$ be the set of edges brought by $\mathcal{A}$ to the central coordinator from server $v$ on input $G_\sigma$; moreover, let $E_\sigma' = \bigcup_{v \in [m]} E_\sigma^v$ be the overall set of edges brought to the central coordinator.

Let $\sigma_0 : [k] \to [k]$ be the identity function, where $\sigma_0(i) = i$ for all $i \in [k]$.

**Claim D.3.** *There are $v^* \in V$ and $u^* \in [k]$ such that $((v^*,1),(u^*,2)) \in E_{\sigma_0} \setminus E_{\sigma_0}'$.*

*Proof.* Since $\mathcal{A}$ is a $(\frac{k-1}{2})$-data algorithm, observe that $E'_{\sigma_0} \leq m \cdot (\frac{k-1}{2}) = \frac{k^2-k}{2}$. Moreover,

$$E_{\sigma_0} = \left|\{((v,1),(u,2)) \in V \times U \mid v \leq \sigma(u)\}\right| = \sum_{v \in [k]} (k-v+1) = \sum_{v \in [k]} v = \frac{k^2+k}{2}.$$

The second equality holds since $\sigma_0 : [k] \to [k]$ is a bijection; thus, for each $v \in [k]$ there are $(k-v+1)$ entries $u \in [k]$ such that $v \leq \sigma_0(u)$. By the above

$$E_{\sigma_0} = \frac{k^2+k}{2} > \frac{k^2}{2} > \frac{k^2-k}{2} \geq E'_{\sigma_0}$$

This gives the statement of the claim. $\lrcorner$

Let $v^* \in V$ and $u^* \in [k]$ such that $((v^*,1),(u^*,2)) \in E_{\sigma_0} \setminus E'_{\sigma_0}$ as promised by Claim D.3. Let $\sigma^* : [k] \to [k]$ be the bijection defined for all $i \in [k]$ as

$$\sigma^*(i) = \begin{cases} u^*, & i = v^* \\ v^*, & i = u^* \\ i, & \text{otherwise} \end{cases}$$

Clearly, $\sigma^*$ is a one-to-one mapping (a bijection). By the definition of $P_{\sigma^*}^{v^*}, P_{\sigma_0}^{v^*}$, recall that $((v^*,1),(u^*,2)) \in P_{\sigma^*}^{v^*}$ and $((v^*,1),(u^*,2)) \in P_{\sigma_0}^{v^*}$.

**Claim D.4.** $E_{\sigma^*}^{v^*} = E_{\sigma_0}^{v^*}$.

*Proof.* Since $\mathcal{A}$ is a zero communication algorithm, $E_{\sigma^*}^{v^*}, E_{\sigma_0}^{v^*}$ depend only on the algorithm $\mathcal{A}$, and the edge sets of the servers $P_{\sigma^*}^{v^*}, P_{\sigma_0}^{v^*}$, respectively. Observe that

$$\begin{aligned}
P_{\sigma^*}^{v^*} &= \{((x,1),(y,2)) \in E_{\sigma^*} \mid x = v^*\} \\
&= \{((v^*,1),(u,2)) \in V \times U \mid v^* \leq \sigma^*(u)\} \\
&= \{((v^*,1),(u,2)) \in V \times U \mid v^* \leq \sigma_0(u)\} \\
&= \{((x,1),(y,2)) \in E_{\sigma_0} \mid x = v^*\} \\
&= P_{\sigma_0}^{v^*}
\end{aligned}$$

$\lrcorner$

The third equality holds since $\sigma^*(u^*) = v^*$, $\sigma^*(v^*) = u^* \geq v^*$ and for every $x \in [k] \setminus \{u^*, v^*\}$ it holds that $\sigma^*(x) = \sigma_0(x)$; thus, for every $u \in [k]$ it holds that $v^* \leq \sigma^*(u)$ if and only if $v^* \leq \sigma_0(u)$. Since $((v^*,1),(u^*,2)) \in E_{\sigma_0} \setminus E'_{\sigma_0}$, by Claim D.1 and Claim D.2, the only maximum cardinality matching in $G_{\sigma^*}$ does not belong to $2^{E_{\sigma_0}}$, and also to $2^{E_{\sigma^*}}$ by Claim D.4. This is a contradiction that $\mathcal{A}$ is an unweighted MWM algorithm with zero communication that in particular finds a maximum matching in $G_{\sigma^*}$. $\square$

## D.2 Lower Bound for Weighted Instances

**Theorem 1.4.** *For every $k \in \mathbb{N}$, there is no randomized $o(k^2)$-data algorithm for $k$-MWM.*

*Proof.* Let $m \in \mathbb{N}$ be the number of servers and let $k \in \mathbb{N}$ be the solution size parameter. Assume towards a contradiction that there is a $\left(\frac{k^2}{2} - 1\right)$-data algorithm $\mathcal{A}$ that solves MWM with zero communication. For every $s \in [m]$, let $B_s = (U, V_s, E_s)$ be a complete bipartite graph with $k$ vertices on each bipartition; namely, let $U = \{u_1, \ldots, u_k\}$, $V_s = \{v_1^s, \ldots, v_k^s\}$ be disjoint sets of vertices of cardinality $k$ and let $E_s = U \times V_s$. Let $W \in \mathbb{R}_{>0}$ be the weight of all edges in $E_s$. Fix some $s \in [m]$ and assume that server $s$ received $E_s$ (with weight $W$ for each edge) as an input, as part of a D-MWM problem (note that the input to the remaining servers can be defined in various ways resulting in different instances). Since $\mathcal{A}$ is an algorithm for MWM with zero communication, it sends a random subset of edges $E^s \subseteq E_s$ to the central coordinator, where $E^s$ depends only on $E_s, W$, and the random bits of the algorithm; that is, it is independent of the input to the remaining servers.

**Claim D.5.** *For every $s \in [m]$ and $e \in E_s$ it holds that $\Pr\left(e \in E^s\right) \geq \frac{1}{2}$.*

*Proof.* Fix some $s \in [m]$. Assume towards a contradiction that there is $e = (u_i, v_j^s) \in E_s$ such that $\Pr\left(e \in E^s\right) < \frac{1}{2}$. Define bipartite graphs $B_1 = (U \setminus u_i, X, E_1)$, $B_2^s = (V_s \setminus v_j^s, Y, E_2^s)$, where $X = \{x_1, \ldots, x_k\}$, $Y = \{y_1, \ldots, y_k\}$, $U, V_s$ are disjoint sets of vertices, $E_1 = \{(u_\ell, x_\ell) \mid \ell \in [k] \setminus \{i\}\}$, and finally $E_2^s = \{(v_\ell^s, y_\ell) \mid \ell \in [k] \setminus \{j\}\}$. Let $\mathcal{V}_s = U \cup V_s \cup X \cup Y$ and $\mathcal{E}_s = E_s \cup E_1 \cup E_2^s$. Let the weight of each edge in $E_1 \cup E_2$ be $2 \cdot W$. Let $G_s = (\mathcal{V}_s, \mathcal{E}_s)$ be an MWM instance $I$, with the distribution $E_s, E_1 \cup E_2^s$ into two servers, and an empty set is given as input for the remaining $m - 2$ servers. Let $M_s = E_1 \cup E_2^s \cup \{(u_i, v_j^s)\}$. Observe that each $\mu \in \mathcal{V}_s$ appears as an endpoint of exactly one edge in $M_s$; hence, $M_s$ is a perfect matching in $G$. Moreover, $E_1 \cup E_2 \subseteq M_s$, and $E_1 \cup E_2^s$ are the edges of strictly the highest weight in $\mathcal{E}_s$; therefore, $M_s$ is the unique maximum weight matching in $G_s$. Hence, as $(u_i, v_j^s) \notin E^s$ with probability at least $\frac{1}{2}$ and $e = (u_i, v_j) \in M$, the central coordinator cannot return an optimal solution for instance $I$ with probability at least $\frac{1}{2}$. We reach a contradiction that $\mathcal{A}$ is required to solve every MWM instance optimally with probability at least $\frac{1}{2}$. ⌋

Let $G = (V, E)$ be the graph where $V = U \cup \bigcup_{s \in [m]} V^s$ and $E = \bigcup_{s \in [m]} E_s$. Consider the MWM instance with the graph $G = (V, E)$ and partition $E_1, \ldots, E_m$ into the $m$ servers, where the weight of each edge in $E$ is $W$. Call this MWM instance $J$. By Claim D.5, it follows that for every $s \in [m]$ and every $e \in E_s$ it holds that $\Pr\left(e \in E^s\right) \geq \frac{1}{2}$. Therefore, overall the expected number of edges sent to the coordinator on instance $J$ using algorithm $\mathcal{A}$ is at least $\frac{m \cdot k^2}{2}$. Note that the cardinality of a maximum matching in $G$ is at most $k$ since every edge in $E$ is connected to a vertex in $U$ and $|U| = k$. Thus, it follows that $\mathcal{A}$ is a $\frac{k^2}{2}$, in contradiction that $\mathcal{A}$ is assumed to be a $\left(\frac{k^2}{2} - 1\right)$-data algorithm. $\square$

# E  Parallel Algorithms

We start with some definitions before giving the proof of Theorem 5.1. Recall that in a *subset selection* problem we are given a set of elements $E$, a weight function $w : E \to \mathbb{R}$ and there is a set $I \subseteq 2^E$ of feasible solutions, which is not necessarily given; the goal is to find $S \in I$ of maximum/minimum $w(S)$. Clearly, maximum weight matching and matroid independent set of maximum weight are natural subset selection problems. As before, $k$ is used to denote the *solution size*. Assuming the solution size is given in the input, we denote by $I = (E, w, k)$ an instance of a subset selection problem.

**Theorem 5.1.** *Let $\mathcal{P}$ be a subset selection problem that has a zero communication $f(k)$-data algorithm $\mathcal{A}$ in time $\tilde{O}(g(|E_i|))$ for each server $i \in [m]$, for some monotonic polynomial functions $f, g$, where $k$ is the solution size. Then, there is an $\tilde{O}\left(g\left(f(k)\right)\right)$ time parallel algorithm that given an instance of $\mathcal{P}$ returns a subset of the elements of size $\tilde{O}(f(k))$ that contain an optimal solution. The algorithm is deterministic if $\mathcal{A}$ is deterministic.*

*Proof.* Let $\mathcal{P}$ be a subset selection problem and let $\mathcal{A}$ be a zero communication $f(k)$-data algorithm in time $\tilde{O}\left(g\left(|E_i|\right)\right)$ for each server $i$, for some monotonic polynomial functions $f, g$, where $k$ is the solution size. Assume without the loss of generality that $f(k), g(k) \geq 1$ and $k \geq 1$. Let $I = (E, w, k)$ denote an instance of $\mathcal{P}$, where $E$ is a set of elements, $w$ is a weight function, and $k$ is the solution size. Define the following algorithm $\mathcal{B}$ on instance $I$. Let $n = |E|$.

1. Let $j \leftarrow 0$, $E^0 \leftarrow E$, $n_0 \leftarrow n$

2. **While** $\frac{n_j}{4 \cdot f(k)} > 1$:

   (a) Let $m_j = \left\lceil \frac{n_j}{4 \cdot f(k)} \right\rceil$

   (b) Partition $E^j$ into servers $E_1^j, \ldots, E_m^j$ where $\left|E_i^j\right| \leq 4 \cdot f(k)$ for all $i \in [m_j]$

   (c) Execute $\mathcal{A}$ on the instance $I_j = \left\{(E_i^j, w, k)\right\}_{i \in [m_j]}$.

(d) Let $K_i^j \subseteq E_i^j$ be the elements brought from server $i$ to the central coordinator $\forall i \in [m_j]$

(e) Update $E^{j+1} \leftarrow \bigcup_{i \in [m_j]} K_i^j, n_{j+1} \leftarrow |E^{j+1}|, j \leftarrow j+1$

3. Return $E^j$.

**Claim E.1.** *The running time of $\mathcal{B}$ on instance $I$ is $\tilde{O}(g(f(k)))$.*

*Proof.* We show two things. First, the number of iterations of the **while** loop is bounded by $O(\log n)$; that is, the value of $j$ at the end of the algorithm satisfies $j = O(\log n)$. Second, each iterations runs in parallel time of $\tilde{O}(g(f(k)))$. Fix some iteration $j$ of the **while** loop such that $n_j > 4$. Observe that $\left|K_i^j\right| \leq f(k)$ since $\mathcal{A}$ is an $f(k)$-data algorithm; thus,

$$
\begin{aligned}
|E^{j+1}| &= \left| \bigcup_{i \in [m_j]} K_i^j \right| \\
&\leq \sum_{i \in [m_j]} \left|K_i^j\right| \\
&\leq f(k) \cdot m_j \\
&= f(k) \cdot \left\lceil \frac{n_j}{4 \cdot f(k)} \right\rceil \\
&\leq f(k) \cdot \frac{n_j}{4 \cdot f(k)} + 1 \\
&< f(k) \cdot \frac{n_j}{4 \cdot f(k)} + \frac{n_j}{4} \\
&= \frac{n_j}{4} + \frac{n_j}{4} \\
&= \frac{n_j}{2}
\end{aligned}
$$

Thus, in every iteration $j$ of the **while** loop besides the last iteration (only in the last iteration it may hold that $n_j \leq 4 \leq 4 \cdot f(k)$), the number of elements decrease by at least half. We conclude that the number of iterations is therefore bounded by $O(\log n)$. Moreover, in each iteration $j$ we can partition the elements into $m_j$ consecutive segments, each containing $4 \cdot f(k)$ elements besides perhaps the last segment which may contain a smaller number of elements. Therefore, by the definition of $\mathcal{A}$ the running time of each iterations takes

$$
\max_{i \in [m_j]} \tilde{O}\left(g\left(E_i^j\right)\right) = \tilde{O}(g(4 \cdot f(k))) = \tilde{O}(g(f(k)))
$$

The equalities above rely on $f, g$ being monotonic polynomial. Thus, overall the running time is bounded by $\tilde{O}(g(f(k)))$. ⌟

**Claim E.2.** *Algorithm $\mathcal{B}$ returns an optimal solution for $I$.*

*Proof.* We prove an invariant of the algorithm, that is preserved in all iterations. The invariant is that for every iteration $j$ of the **while** loop of the algorithm, there is an optimal solution $S$ of $I$ such that $S \subseteq I_j$. Before the first iteration, it holds that $E^j = E$ and the invariant is trivially satisfied. Assume that the invariant holds before iteration $j$; thus, there is an optimal solution $S$ of $I$ such that $S \subseteq E^j$. Then, in iteration $j$ we apply $\mathcal{A}$ on $I_j$; since $\mathcal{A}$ is a zero communication algorithm for $\mathcal{P}$ it follows that there is an optimal solution $S \subseteq E^{j+1}$ for $I_j$. Since there is an optimal solution for $I$ that is a subset of $E^j$, we conclude that an optimal solution for $I_j$ is also an optimal solution for $I$. Thus, there is $S \subseteq E^{j+1}$ that is an optimal solution for $I$. Overall, it follows that there is an optimal solution that is a subset of the elements brought to the central coordinator. This implies that $\mathcal{A}$ returns a subset of elements containing an optimal solution for $I$ in this last iteration and consequently $\mathcal{B}$ returns a subset of elements containing an optimal solution for $I$ as required. ⌟

The proof of the theorem follows from Claim E.1 and Claim E.2 since at the end of the algorithm it holds that $|E^j| = O(f(k))$. □

We give below the proofs for our parallel algorithms for $k$-MWM and MWM. We will use the following algorithm for $k$-MWM.

**Lemma E.3.** [25] *There is a deterministic algorithm that solves $k$-MWM in time $\tilde{O}\left(k \cdot (|V| + |E|)\right)$.*

The algorithm used for the above lemma by Gabow [25] is a variation of the celebrated algorithm of Edmonds [21] for MWM (a generalization of the Hungarian algorithm [39]). Specifically, this algorithm is made of at most $|E|$ *searches* – finding an augmenting path in the graph. Each path augments by one the cardinality of the current matching, and is of maximum weight among all matchings of the current cardinality. Thus, after $k$ searches, the algorithm solves $k$-MWM. As Gabow [25] constructs an implementation of a single search of this algorithm in time $\tilde{O}\left(|V| + |E|\right)$, overall $k$ searches give the running time given in the above lemma. We note that the running time of the lemma may be improved using an adaptation of faster algorithms for special cases of MWM (e.g., smaller weights [26] or bipartite graphs [55, 39]). However, this requires an adaptation of such an algorithm to $k$-MWM instead of MWM, which is not always applicable.

**Theorem 1.7.** *There is a deterministic parallel $\tilde{O}\left(k^3\right)$ time algorithm for $k$–MWM.*

*Proof.* By Theorem 1.2, there is a deterministic zero communication $O\left(k^2\right)$-data algorithm for maximum weight matching in a deterministic parallel time of $\tilde{O}\left(|E_i|\right)$ for each server $i$. As each edge and its weight can be encoded in $\tilde{O}(1)$ bits, each server sends $\tilde{O}\left(k^2\right)$ bits overall. Thus, by Theorem 5.1 (using the above, i.e., $f(k) = O\left(k^2\right)$ and $g\left(|E_i|\right) = \tilde{O}\left(|E_i|\right)$), there is a parallel deterministic $\tilde{O}(k)$ algorithm that returns a subset $S$ of $\tilde{O}\left(k^2\right)$ edges containing an optimal solution. Let $G' = (V', E' = S)$ be the induced graph on these edges. Note that $|E'| = |S| = \tilde{O}\left(k^2\right)$; therefore, $|V'| \leq |E'| = \tilde{O}\left(k^2\right)$. Then, using Lemma E.3, we can solve this residual $k$-MWM instance in deterministic time $\tilde{O}\left(k \cdot (|V'| + |E'|)\right) = \tilde{O}\left(k \cdot \left(k^2 + k^2\right)\right) = \tilde{O}\left(k^3\right)$. $\qquad\qquad\square$

We complete this section with our algorithm for MWM, which uses the above algorithm.

**Theorem 1.8.** *There is a deterministic parallel $\tilde{O}\left(k^4\right)$ time algorithm for* MWM.

*Proof.* In this proof, we use the algorithm given in Theorem 1.7 to solve MWM (rather than $k$-MWM); that is, without the cardinality bound. The approach uses the above algorithm, denote it by $\mathcal{A}$, as a black box, using increasing values of $k$. Specifically, we run the following algorithm $\mathcal{B}$ based on algorithm $\mathcal{A}$ (defined in Theorem 1.7) for $k$-MWM. Assume without the loss of generality that the graph contains at least one edge with strictly positive weight (otherwise the problem is trivial).

1. Initialize $k = 0$

2. Update $k \leftarrow k + 1$

3. Execute $\mathcal{A}$ on the input graph with parameter $k$; let $w_k$ be the total weight of the solution

4. **If** $w_k \leq \max_{q \in \{\lfloor k/3 \rfloor, \ldots, k-1\}} w_q$:

    (a) **return** the highest weight solution found over all iterations $q = 1, \ldots, k$

5. **else**: Go to Step 2

Let $k^*$, opt be the minimum cardinality of an optimal solution for the given instance and such an optimal solution, respectively. By the minimality of opt, all edges in opt have strictly positive weight. Then, during the execution of $\mathcal{B}$, note that for every $k < k^*$ it holds that $w_k > \max_{q \in \{\lfloor k/3 \rfloor, \ldots, k-1\}} w_q$ since there is an edge in opt with strictly positive weight that can be added to the solution of iteration $\lfloor k/3 \rfloor$ (for every two matchings $A, B$, where $|A| < 2 \cdot |B|$ there is an edge in $B$ that can be added to $A$ preserving the matching property, as each vertex is an endpoint of at most one edge in a matching). Thus, the condition in Step 4 fails for every $k < k^*$; consequently, during the execution of $\mathcal{B}$, we execute $\mathcal{A}$ with cardinality $k^*$ in particular, which guarantees that $\mathcal{B}$ returns an MWM.

It remains to bound the running time. Each iteration in the execution of algorithm $\mathcal{A}$ with cardinality $k$ takes $\tilde{O}\left(k^3\right)$ deterministic parallel time using Theorem 1.7. Hence, to conclude, we show that the number of iterations of executing $\mathcal{A}$ is $O\left(k^*\right)$; thus, as in all of these iterations it holds that $k = O\left(k^*\right)$ the proof follows. Let $k'$ be the value of $k$ at the end of the algorithm, and it follows that $w_{k'} \leq \max_{q \in \{\lfloor k'/3 \rfloor, \ldots, k'-1\}} w_q$ by the stopping condition of the algorithm. Note that for every $k$ such that $\lfloor k/3 \rfloor \geq k^*$ it holds that $w_k \leq \max_{q \in \{\lfloor k/3 \rfloor, \ldots, k-1\}} w_q$ since opt is a MWM (and $\mathcal{A}$ returns a $k^*$-MWM in iteration $k^*$, which has a weight equals to the optimal weight). Thus, $k' = O\left(k^*\right)$ as required. $\qquad\square$

# F   Lower Bound for Shortest Paths

In this section, we show that there is no algorithm that solves the unweighted shortest path problem even for paths of constant length without bringing the entire set of edges to the central coordinator. Intuitively, this differentiates the parameterized zero communication algorithms from FPT algorithms, using the same parameters, since shortest path can be solved in polynomial time and in particular has an FPT algorithm parameterized by the solution size.

**Theorem F.1.** *There is no $d$-data algorithm for distributed shortest path for any $d = o(|E_i|)$, even for paths of length $O(1)$, where $E_i$ is the set of edges of server $i$.*

*Proof.* Assume towards a contradiction that there is a zero communication $d$-data algorithm $\mathcal{A}$ for $d < |E|$, where $E_i$ is the set of edges of server $i$. Consider the following instance. Let $n \in \mathbb{N}$ be a sufficiently large integer and let $U = \{u_1, \ldots, u_n\}$, $V = \{v_1, \ldots, v_n\}$, $W = \{w_1, \ldots, w_n\}$ be disjoint sets of vertices. Let $E_{UV} = \{(u_i, v_i)\}_{i \in [n]}$ and $E_{VW} \subseteq \{(v_i, w_i)\}_{i \in [n]}$ be sets of edges between $U, V$ and $V, W$, respectively. Let $V = U \cup V \cup W \cup \{s, t\}$ where $s, t \notin U \cup V \cup W$ and let Let $E = E_{UV} \cup E_{VW} \cup (\{s\} \times U) \cup (\{W\} \times \{t\})$. Consider a collection of shortest path instances on 2 servers (and a central coordinator), where the edges are distributed between the server by $E_{UV} \cup (\{s\} \times U)$ in server 1 and $E_{VW} \cup (W \times \{t\})$ in server 2. Note that this is a collection of instances since we did not describe $E_{VW}$ explicitly and only require $E_{VW} \subseteq \{(v_i, w_i)\}_{i \in [n]}$. Since $d = o(|E_{UV}|)$ and $E_{UV} = \Omega(n)$ and as $n$ is sufficiently large, there is $(u_i, v_i) \in E_{UV}$ not brought to the central coordinator from Server 1 in an execution of $\mathcal{A}$ on the instance $G = (V, E)$. Since $\mathcal{A}$ is a one-round algorithm, $(u_i, v_i)$ is independent of the set of edges in Server 2, in particular on $E_{VW}$. Now, consider the case where $E_{VW} = \{(u_i, v_i)\}$. Then, in this case the only path in $G$ between the source $s$ to the target $t$ is $(s, u_i), (u_i, v_i), (v_i, w_i), (w_i, t)$. However, since $(u_i, v_i)$ is not brought to the central coordinator by $\mathcal{A}$ the algorithm returns that there is no path between $s$ and $t$, in contradiction to the fact that $\mathcal{A}$ is an algorithm that requires to solve all shortest path instances optimally. We conclude that such an algorithm $\mathcal{A}$ does not exist. $\qquad\square$

