# OpenReview forum: "Tight Bounds for Maximum Weight Matroid Independent Set and Matching in the Zero Communication Model"
_NeurIPS.cc/2025/Conference — NeurIPS 2025 poster_

### Official Review · Reviewer_qVML · 2025-06-30

**Clarity:** 3
**Significance:** 3
**Originality:** 4
**Rating:** 4
**Confidence:** 2

**Summary:**

This paper investigates the problem of computing exact solutions for the Maximum Weight Matroid Independent Set (MW-IS) and the Maximum Weight Matching (MWM) problems under a strict zero-communication mode. Each of the m distributed servers processes part of the data and then transmits a message to the central coordinator without any inter-server coordination. The authors provide deterministic algorithms that solve this problem with tight bounds: O(k) per server data for the MW-IS problem and O(k^2) for the MWM problem where k is the solution size. This paper also presents matching lower bounds. The techniques proposed in this paper are further leveraged to derive deterministic parallel algorithms for both of the discussed problems (MW-IS and MWM).

**Questions:**

1. What happens in the presence of noisy or faulty server inputs? Do the bounds proposed in this paper still hold?
2. Have the authors tested any of these methods in practice? If so, could they comment on how these deterministic algorithms compare in runtime to the randomized ones discussed in the paper?

**Ethical Concerns:**

["NO or VERY MINOR ethics concerns only"]

**Final Justification:**

The paper makes a clear theoretical contribution in a specialized zero-communication model. The authors rebuttal helped clarify the motivation and addressed some open questions, including possible extensions and the impact of noise. However, the lack of empirical results and limited discussion of real-world relevance remain as minor weaknesses. I still find the work interesting and well-executed on the theoretical side, and I am keeping my original score.

**Limitations:**

Yes

**Quality:**

3

**Strengths And Weaknesses:**

Strengths:
- Good theoretical contributions with rigorous proofs and tight bounds for both the algorithm results and the lower bounds.
- Relevant exploration of a severely constrained communication model.
- Demonstrates applicability in parallel computing by linking distributed computation to efficient parallelism.

Weaknesses:
- There weren't any experimental results provided but this is a soft weakness since the authors may think it's not necessary.
- The initial setting of this paper (zero-communication exact algorithms) is a fairly specialized setting so applicability to the real-world is only lightly motivated. Perhaps it could be further motivated.

---

> ### Author Rebuttal · Authors · 2025-07-30
>
> We thank the reviewer for their helpful comments and suggestions. We answer their questions below.
>
>
>
>
> - There weren't any experimental results provided, but this is a soft weakness since the authors may think it's not necessary.
>
>
> Answer: We thank the reviewer for this remark. We believe the contribution of this paper is foremost theoretical. However, if the reviewer thinks the paper would benefit from empirical experiments, we would be happy to consider adding such experiments to the full version of the paper.
>
> - The initial setting of this paper (zero-communication exact algorithms) is a fairly specialized setting, so applicability to the real world is only lightly motivated. Perhaps it could be further motivated.
>
> Answer: We agree with the reviewer that we should elaborate more on motivating the model and will do so in the ready version of the paper. We believe the zero-communication model is worth studying for several key reasons. First, as mentioned by the reviewer, we showed its close connection to parallel algorithms. Second, it allows a clean setting to distinguish between the difficulty of problems in a common distributed setting. We can think of problems that fit well in this model as “communication-efficient” – that can be distributed with small/no communication overhead. Knowing which problems exhibit this property can have practical implications and enable robust algorithms that do not have to process large data on a single processor. Thus, “easy” problems like shortest paths in a classic computational model that are hard in this model show the usefulness of the model in distinguishing problems according to a restrictive model of communication complexity. Finally, the model generalizes well-studied problems in the literature, such as one-direction communication, making it a natural extension of prior work (as discussed in the introduction).
>
>
> Questions of the reviewer:
>
> 1. What happens in the presence of noisy or faulty server inputs? Do the bounds proposed in this paper still hold?
>
> Answer: This is a very interesting question. The noise can be either on (1) the actual element – whether it is given to the server or not, (2) the feasibility – the server may not know exactly which subsets of elements are feasible, and (3) the weights can be noisy. This of course, only makes the problem more difficult, and our lower bounds hold, and the question is if the algorithms given in the paper can be generalized to tackle any of these scenarios. While the answer depends on the exact formulation of the noise model, our bounds are unlikely to hold as is now; it remains an interesting question for follow-up research to rigorously define a variation of our model on faulty servers and give tight bounds for the discussed problems (and more) in such a model. We will add this to the discussion.
>
> 2. Have the authors tested any of these methods in practice? If so, could they comment on how these deterministic algorithms compare in runtime to the randomized ones discussed in the paper?
>
> Answer: As mentioned above, we did not test our results empirically, but we can do so if the reviewer finds it useful. We naturally expect that for small values of k, our parallel running time bounds will be similar to the randomized ones, but degraded for larger values of k approaching n. On the other hand, these randomized algorithms are algebraic and may have practical overheads in contrast to our relatively simple deterministic approach.

---

### Official Review · Reviewer_5oXm · 2025-07-01

**Clarity:** 4
**Significance:** 2
**Originality:** 2
**Rating:** 4
**Confidence:** 4

**Summary:**

This paper is interested in the problem of selecting a maximum weight feasible set (independent set in a matroid, matching, etc) using $m$ servers which may only communicate through a central coordinator. The motivation is that perhaps the dataset is huge and needs to be partitioned onto $m$ servers -- what should the servers send the central coordinator to guarantee that the central coordinator can output the right answer? How much data do the servers have to send? How much computation do they individually have to do? The paper gives upper and lower bounds to these questions.

**Questions:**

Can your results with matroids extend to matroid intersection in any way? That might imply some (bipartite) matching results. I would be interested in knowing that.

You should cite coresets in your paper. You're basically creating coresets. This is a rich body of work, which may have implications about some of your questions / results.

What's the difference between $d$-data and the definition of bandwidth?

**Ethical Concerns:**

["NO or VERY MINOR ethics concerns only"]

**Final Justification:**

The original positive review I gave for this paper stands. The main thing I would definitely want from this paper if accepted is something to address this paragraph from my review:

"My main worry is that Section 4's results seem the most interesting to me, but not enough details of that proof have been added to the main body to convince me the result is true. I think this is critical in a theory focused paper (even though there's only 9 pages the reviewers will be reading). _Note: if you can write more about why your more non-trivial results are true and how you went about proving it, I would be more likely to bump my assessment to a 5_. "

It's not so hard hitting I would make it a strong accept (I would want really interesting proofs and/or experiments for that).

**Limitations:**

Yes.

**Paper Formatting Concerns:**

I did not notice any major formatting issues.

**Quality:**

3

**Strengths And Weaknesses:**

Weaknesses:

Section 3's proof is a little easy. But there's sort of a subculture within theory to de-value papers just because the proofs seem easy, so I don't think that's good grounds to reject a paper. Still, you devote almost two pages of valuable space to a proof that really just needs a few paragraphs. I'm much more interested in the proofs of Section 4's results (which are not in the main body, but the appendix). Section 4 proofs should be taking the pages you've devoted to Section 3. For example, I would really like to understand more intuition in Definition 4.1. I see how properties (1) and (3) follow from the algorithm in lines 331 - 337. Can you say a few lines about why property (2) holds (this should be in the main body, not appendices)? Also, include a paragraph as to why the union of strong sets must contain a maximum weight matching.

My main worry is that Section 4's results seem the most interesting to me, but not enough details of that proof have been added to the main body to convince me the result is true. I think this is critical in a theory focused paper (even though there's only 9 pages the reviewers will be reading). _Note: if you can write more about why your more non-trivial results are true and how you went about proving it, I would be more likely to bump my assessment to a 5_.

This is a nice theoretical result, and I'm not super sure what it's doing here in NeurIPS over some other theory conference. The application is quite nice. There's no evidence / results about empirical performance though, which I usually like to see in a theory focused paper in a venue like this.

Strengths:

This is a great model. The proofs I read are clean and correct. The writing is wonderful and pleasant to read. If NeurIPS is interested in accepting more theory like papers, this is a pretty good candidate. Nice job!

---

> ### Author Rebuttal · Authors · 2025-07-30
>
> We thank the reviewer for their helpful comments and suggestions. We answer their questions below.
>
>
>
>
> - ….For example, I would really like to understand more intuition in Definition 4.1. I see how properties (1) and (3) follow from the algorithm in lines 331 - 337. Can you say a few lines about why property (2) holds (this should be in the main body, not appendices)?
>
>
> Answer: Intuition on Algorithm Strong-Set (for a formal proof read Appendix A):
> For the following, consider an execution of the algorithm on a set of edges E with weight function w. The algorithm returns a subset S of edges, and for this subset to be a strong set, consider some edge e = (u,v) in E and let us give the intuition why necessarily at least one property from (1), (2), or (3) holds for e. We consider several cases here.
>
>  -  First, clearly if e is in S, then property (1) holds for e = (u,v).
> - Second, assume that (1) does not hold for e and we shall show why (2) or (3) must hold. Since e is not added to S, by Step (b) of the algorithm we can consider two sub-cases to conclude this intuitive proof:
>
> I. N(S,u) > 2k or N(S,v) > 2k, where recall that N(S,u) and N(S,v) are the number of neighbors of u and v in S respectively. From this case, (3) easily holds since we have taken to S at least 2k+1 neighbors of x in {u,v} and all of these neighbors are with a higher or equal weight due to the order the edges are iterated.
>
>
> II.  The complementary case, where N(S,u) \leq 2 k and N(S,v) \leq 2*k. Since the algorithm did not take the edge e to S, the reason has to be since it terminated by the stopping condition of Step (c), i.e., we get that |S| > 10 k^2.  Note that by Step (b) of the algorithm, we do not take edges that increase the degree of any vertex over 2 k+1. That is, for each edge e_i = (u_i, v_i) we have N(S,u_i) \leq 2 k + 1 and N(S,v_i) \leq 2 k + 1. Therefore, the edges in S induce a graph with (sufficiently many) over 10 k^2 edges that has a maximum degree of 2 k+1. By a result of Han (reference [20] in our paper, or Claim A.3 to be precise), since there are relatively many edges and the maximum degree is fairly small, there exist a matching in this induced graph of a size of over 2k.
>
>
>
>
>
>
> - Also, include a paragraph as to why the union of strong sets must contain a maximum weight matching.
>
> Answer: We agree with the reviewer that more intuition for strong sets is required. We will add a variation of the following intuition to Section 4.
>
>
>  Intuition for strong-sets (or, intuition for how to use strong sets to prove Lemma 4.3): In Lemma 4.3, we are given a strong set S_i for each server i and need to prove that the union of S_i’s contains a maximum weight matching (MWM). In the proof of Lemma 4.3, we consider an MWM denoted by M^ that has the maximum size of intersection with the union of S_i’s.
>
> If M^ is contained in the union of S_i’s - we are done as the central server can compute M^. Otherwise, there is some element e in M^ and in E_i, for some server i, that does not belong to S_i. Here, we use the fact that S_i is a strong set to find some e^ in S_i such that M’ = M^ \setminus {e} \cup {e^} is an MWM. Specifically, recall the definition of a strong set S_i and the edge e, which guarantees one of the following (Definition 4.1):
>
> e is in S_i - we already know this is not the case by the definition of e.
>
> There is a matching M contained in S_i of (a sufficiently large) cardinality 2k+1 such that every edge in M has higher or equal weight to the weight of e. Thus, as M is sufficiently large and since M and M^ are matchings, there is e^ in M such that M’ = M^ \cup {e^} is a matching (more specifically, since M^ and M are matchings, each endpoint of M^ can appear at most once as an endpoint of M, and since |M| > 2 |M^| there is an edge in M that can be added to M^ resulting in a matching). Finally, the weight of M’ can only be higher or equal to the weight of M^ since e^ in M.
>
> There is an endpoint x of e and there are 2k+1 distinct edges adjacent to x: (x, v1), . . . ,(x, v_{2 k+1}), each of weight at least the weight of e. Then, in this case, we can use one of these edges as e^ to define M’ = M^* \setminus {e} \cup {e^} analogous to the above case. In more detail, since |M^| \leq k it has at most 2k endpoints. Therefore, as there are 2k+1 edges as mentioned above, one of these edges e^=(x,v_j) satisfies that v_j is not an endpoint of M^; thus, if we remove e from M^ and add e^, it would induce a matching M’ = M^ \setminus {e} \cup {e^} (as the endpoint x replaces its matched endpoint to a new endpoint not in M^), that will have higher or equal weight to the weight of M^.
> The above cases yield a contradiction that M^ is an MWM of maximum intersection with the union of S_i’s.
>
>
>
>
>
>
>
>
>
>
>
>
>
>
>
>
> - My main worry is that Section 4's results seem the most interesting to me, but not enough details of that proof have been added to the main body to convince me the result is true. I think this is critical in a theory focused paper (even though there's only 9 pages the reviewers will be reading). Note: if you can write more about why your more non-trivial results are true and how you went about proving it, I would be more likely to bump my assessment to a 5.
>
> Answer: We agree with the reviewer on the need to add more intuition to section 4. We will add a slightly shorter description of the above proof ideas of Lemma 4.2 and Lemma 4.3 to the first 9 pages of the paper.
>
> - …There's no evidence/results about empirical performance though, which I usually like to see in a theory-focused paper in a venue like this.
>
> Answer: We believe the contribution of this paper is foremost theoretical. However, if the reviewer think the paper would benefit from empirical experiments, we would be happy to consider adding such experiments to the full version of the paper.
>
>
>
>
> Questions of the reviewer:
>
> - Can your results with matroids extend to matroid intersection in any way? That might imply some (bipartite) matching results. I would be interested in knowing that.
>
> Answer: This is a great question that can be the subject of follow-up research. Note that bןpartite matching is both a special case of matching in a general graph and of matroid intersection. Hence, our zero-communication algorithm for MWM applies to bipartite matching, and some of our lower bounds (e.g., our lower bound for matching on unweighted instances) applies for bipartite graphs.
> We do have an idea for an f(k)-data algorithm for matroid intersection, for an exponential function f. It would be very interesting to solve this for a polynomial function, or show a lower bound ruling this out. We will add this question to the discussion.
>
> - You should cite coresets in your paper. You're basically creating coresets. This is a rich body of work, which may have implications about some of your questions / results.
>
> Answer: We thank the reviewer for this idea and will add a discussion on the relation of our results (specifically on matching) to coresets.
>
> - What's the difference between d-data and the definition of bandwidth?
>
> Answer: We use the term bandwidth to refer to the maximum number of elements delivered to the coordinator from a single server (perhaps this is somewhat confusing and should be termed differently). An algorithm is a d-data algorithm if the bandwidth is at most d, but not necessarily equal to d.

---

> > ### Comment · Reviewer_5oXm · 2025-08-06
> >
> > Thank you for your response, and I appreciate all the changes you've implemented! I think a 4 is an appropriate score for this paper. I think it's a wonderful theoretical result, I just haven't been able to fully verify the matchings result in this short rebuttal time. I maintain my review.

---

### Official Review · Reviewer_SpuD · 2025-07-02

**Clarity:** 3
**Significance:** 2
**Originality:** 3
**Rating:** 4
**Confidence:** 3

**Summary:**

This paper addresses two related problems : Maximum-Weight Matroid Independent Set (MW-IS) and Maximum Weight Matching (MWM) under a 0 communication model, the original problem is partitioned among smaller servers and each server must send a message to a central server such that the union of all messages sent "contains" the optimal solution for the overall problem.  The goal is to minimize the size of the largest message sent.

More specifically, this paper investigates a specialized form of the above problem, parameterized by $k$, which is the "solution size". For example, in the case of MWM, this paper is interested in computing the "k-MWM", that is, the maximum-cost matching of cardinality $k$. Parameterizing based on this $k$, the authors give simple algorithms for finding $k$-MW-IS and $k$-MWM with maximum message sizes of $k$ and $k^2$ respectively. They also provide lower bounds that show these results are optimal.

As a direct implication, the authors also obtain bounds for parallel versions of $O(\sqrt{k}\log{n})$ for $k$-MW-IS and $O(k^4\log{n})$ $k$-MWM.

**Questions:**

I have highlighted several concerns in my discussion above, but, to streamline the response, please specifically address the following, referring to my discussion above for more details:
(1) Please provide practical justification for why one should expect $k$ to be small for MWM, noting that, when $k \geq \sqrt{n}$, your algorithm effectively has no benefit for MWM under either model.
(2) Please provide clarification regarding your parallel MWM algorithm, in light of my discussion above. In partciular:
- Why are there $k^2$ total edges on the central server? (How do you account for the total message size from all servers?).
- What sequential algorithm are you using?
- Can the Hungarian algorithm (or any other faster sequential algorithm, or even another parallel algorithm) yield better results here?

**Ethical Concerns:**

["NO or VERY MINOR ethics concerns only"]

**Final Justification:**

In the rebuttal discussion, my the authors sufficiently addressed most of my concerns. Some of these require relatively minor edits to the paper, which should be easily doable for a final camera ready version. I have increased my rating accordingly.

Authors, please address the following in the updated version:
- Include a more thorough context and discussion of $k$-MWM, both from a theoretical standpoint as well as a practical one.
- Update the discussion of the $k$-MWM parallel algorithm to include the improved bound, as well as some short additional justification for this bound.

**Limitations:**

Nothing to note here.

**Quality:**

2

**Strengths And Weaknesses:**

Strengths : The paper is generally well written, especially with regards to notation definitions, aside from a few things highlighted below. (There is one major concern in the parallel section for MWM that needs addressing).

The inclusion of lower bounds strengthens the theoretical contribution of this paper, especially because it is not clear whether this exact parameterization (based on $k$) has been investigated before under this model.

Weaknesses :
Due to being less familiar with the literature on MW-IS, my feedback here is mainly related to $k$-MWM.

- Your algorithm only makes sense when k is very small. However, why should we expect k to be small? I.e., for what practical applications would k ever be small?
- You mention several applications of, for example, MWM, in a generic sense, and I agree MWM is a very important problem, and solving MWM in a zero communication model is an interesting problem. However, you are solving k-MWM, which is a totally different (probably much easier to solve) problem. As a result, you have to properly motivate *that* specific problem, which I see no discussion of; you only motivate MWM in a somewhat vague / general sense. This is both a practical concern and a theoretical concern, since I don’t think any prior work on the k-MWM variant was cited, and the value of a purely theoretical result is in some sense rooted in its comparison to prior theoretical work.
In particular:
(1)	What are some examples of practical scenarios (for a NeurIPS audience) that motivate the *k*-MWM variant you propose?
(2)	Why should someone expect that this k-MWM variant is a difficult problem to begin with?
(3)	Has the *k*-MWM problem been proposed and studied *outside* of your zero communication scenario? If so, then that would provide a good motivation for the problem to begin with. If not, then why solve it for the zero-communication setting if it’s not interesting in general?
I would like to elaborate by saying that I am aware of the importance of the “partial p-Wasserstein” distance problem, related also to the “partial optimal transport” problem, which are generalizations of (the minimization variant of) your k-MWM problem, and have applications to a machine learning audience. Optimal cardinality-k matchings are important. My concern is that, for these applications, I don’t think k << n. I would normally expect k to be more like Theta(n), or, at least Omega(\sqrt{n}) in most practical cases.
I appreciate the theoretical nature of this work, but I also think that the problem needs stronger practical motivation that relates to a machine learning audience, specifically for motivating that “very small k”.

I think that the literature review is somewhat lacking. The nature of the citations in the related work section seems somewhat disconnected from the nature of this work. The cited related work mentions several results for various communication models, and mainly gives approximation results. However, so far as I can tell, not a single citation in this document addresses the *k*-MWM problem, which is computationally very different from the MWM problem. This absence of prior work cited makes it difficult to assess the theoretical contribution of this work relative to prior work. If nothing else, the Hungarian algorithm is missing, which can compute an optimal cardinality $k$ matching in $O((m + n\log{n}) * k)$ time.

Finally, I don’t find the parallel result for MWM to be particularly surprising, and I don’t think it is at all tight – this appears as more of a direct by-product of your (optimal) 0-communication model, without any attempt at optimizing it for the parallel setting. I believe that prior (old) work for MWM is much stronger for the parallel setting. For example, see:

Orlin, J. B., & Stein, C. (1993). Parallel algorithms for the assignment and minimum-cost flow problems. Operations research letters, 14(4), 181-186.

This also cites some work even prior to that. In this line of work, some bounds are given with parallel running times for the perfect matching problem that have running times of polylogarithmic in $n$ (with perhaps some dependency on the largest edge cost as well). Of course, this is a slightly different problem, since it computes a perfect matching, but in my experience, finding perfect matchings is usually more difficult that finding cardinality $k$ matchings for most combinatorial algorithms.

It is also not clear how your parallel bound for MWM works exactly, as stated. It seems as if you are assuming that the central server ends up with $k^2$ edges, but there are multiple servers sending a message of size $k^2$, right? So shouldn’t the central server end up with a solution much larger than $k^2$, due to accounting for the potentially very large number of servers? Regardless, the way you achieve your parallel bound is collecting a set of edges on the central server and running a (mysteriously unspecified) sequential algorithm. Why run a sequential algorithm instead of some existing parallel algorithm, such as:

Gabow, H., & Tarjan, R. (1988, January). Almost-optimum speed-ups of algorithms for bipartite matching and related problems. In Proceedings of the twentieth annual ACM symposium on Theory of computing (pp. 514-527).
In general, it seems to me like this parallel section for MWM is not thoroughly treated, and this section is perhaps the weakest part of the paper.

Corrections / Improvements :
55 : There are multiple types of randomization. One option is that randomization affects the accuracy, and another is that it affects the efficiency (Monte Carlo or Las Vegas). My point being, it’s possible to have an algorithm that is guaranteed optimal with probability 1, but that terminates within the desired time bound, with, say, probability 1/2. Most would still consider that “randomized”, but it is outside of the definition you’ve specified. This is not a major concern, but please add clarification to your model description to clarify the differences between these probability models, and also double check your proofs (especially the lower bounds) to ensure that they continue to apply under the more general notion of “randomized algorithms” if needed. If you are able to use a more general definition of probability, you should do so. Otherwise, you should clearly emphasize the limitation.

Introduction : Whenever you mention that the edges sent to the coordinator *contain* the optimal solution, be a bit more formal: i.e., say that there exists an optimal solution such that it’s a subset of the provided edges. The reason for this recommendation is that, as written, the reader might wrongly assume that the optimal consists of the optimal rather than being a superset of it. There is nothing incorrect here; simply a matter of readability.

108 : “running time of server i can be computed in “. I don’t think you mean “computing the *running time*”. Reword this.

180 : “In this work, we *showed* tight bounds”  : Due to the placement, this wording seems odd, more like it would be a conclusion. Same again on next line. Showed -> show.
Introduction : It is mentioned in multiple places (line 84 and 86) that k is the “solution size”, which seems to imply the “size of the optimal solution”. However, you really mean that k is a “constraint on the set of feasible solutions”. I.e., in your official problem statements, you are considering the maximum weight solution restricted to subsets of size k. I think some parts of the introduction need to be reworded to clarify this distinction, as I found it confusing. (It was correct and complete, but rewording these occurrences will help readability).

Lines 377 – 378 : Question about your k^4 bound: Where do you get k^4? Which sequential algorithm are you applying? Are you trying to apply the Hungarian algorithm, and assuming it runs in O(m^2) time? There should be a trivial improvement to this: The Hungarian algorithm takes only $O((m + n\log{n})*k)$ running time when computing an optimal cardinality k matching. (Just run the first k iterations of it, using Dijkstra’s algorithm as a Hungarian search). So, your running time should simply be roughly ~O(k^3) I think if there are indeed only $k^2$ many edges. I know this to be true for the minimization problem where all costs are non-negative, but it seems like some small modifications to the edge costs can make the two problems equivalent (maximization for all matchings of size at most k). Please double check on this.

Regardless, you have to state what sequential algorithm you are using. Make it explicit!

---

> ### Author Rebuttal · Authors · 2025-07-30
>
> We thank the reviewer for their helpful comments and suggestions. We start by answering their direct questions that appear at the end of the review, and then comment on the additional questions that came up in the review text.
>
> (1) Please provide practical justification for why one should expect k to be small for MWM
>
> Answer: We agree with the reviewer that the paper does not include a sufficient reference to applications for the regime where k << n. We will add a subsection devoted to that to the paper (including the closely related partial optimal transport problem mentioned by the reviewer).
> The most prominent applications of the maximum matching problems are in bipartite graphs, where we have two sides representing sellers and buyers, tasks and machines, drivers and riders, and many more. A common setting where k is small occurs when the two sides are unbalanced.
>
> For example, consider a server with k processing slots and a given set of n>>k tasks, each task can be scheduled on some of the processors but potentially not all of them due to compatibility issues such as memory communication bandwidth, or other performance or security reasons. This induces a bipartite graph with k vertices on one side and n>>k vertices on the other side, and the goal of the server is to compute a maximum weight matching (the weight being the price a task is willing to pay to be scheduled on a corresponding processor).
>
>
> As a second example, consider the problem of ad allocation in internet browsers. In this setting, when a user searches a string, the browser fills the first k places with ads, where there are n>>k ads that are willing to pay for a slot on this search. Each link between an ad x and slot y creates an edge (x,y) with weight being the bid of x, usually multiplied by a click-rate. The goal of the browser is to allocate ads to the slot, maximizing the total price paid – an MWM instance where k is naturally significantly smaller than n.
>
> Furthermore, the study of parameterized complexity, in particular on the solution size, goes beyond practical applications and is a core research field in theoretical computer science. We believe that understanding the parameterized complexity (in various models) of fundamental problems is an important task.
>
>
>
>
> (2) Please provide clarification regarding your parallel MWM algorithm, in light of my discussion above. In particular:
> Why are there k^2 total edges on the central server? (How do you account for the total message size from all servers?).
>
>
> Answer: At a high level, our parallel algorithm for MWM can be described as follows:
> - While the number of edges is larger than ck^2 for some constant c (a specific value of c can be smaller than, e.g., 30, which follows from our construction in Section 4):
> - We partition the edges (in parallel) evenly and arbitrarily into roughly |E^j| / c k^2 servers, where E^j is the current set of edges (initialized as E, the given set of edges).
> - Execute our O(k^2)-zero-communication algorithm A on the instance induced from the above partition (using the initial given parameter k). The constant c is sufficiently large so our algorithm is less than a (c/2) k^2-data algorithm.
> - Update E^{j+1}, the elements for the next iteration, as the union of elements brought by Algorithm A in the above step to the central server.
> - When the number of remaining edges becomes at most c k^2, we solve this instance using a sequential algorithm (see the comments below on the selection of this algorithm)
>
>
> The intuition is that at each iteration, each server applies filtering of the edges it receives: The server gets roughly ck^2 edges and returns only at most half, i.e., (c/2) k^2 edges (in parallel to the other servers). Thus, in O(log n) such rounds, we remain with an instance with only m’ = O(k^2) edges and therefore only n’ = O(k^2) vertices as well. Note that this is a parallel algorithm and not an algorithm in the zero-communication model, allowing us to exploit in each round a zero-communication algorithm with different number of servers: In the first round, the number of servers is roughly |E| / ck^2; in the second round it is |E| / (c/2) k^2, etc., until after O(log n) such round we remain with O(k^2) edges that can be solved efficiently enough using a standard MWM algorithm (see below).
>
>
>
> - What sequential algorithm are you using?
>
> Answer: As there is an O(mnpolylog(n)) time algorithm for MWM, we solve the last sequential step of our parallel algorithm in time O(m’ n’) = O(k^4polylog(n)) using the following classic result:
>
> Galil, Z., Micali, S., and Gabow, H. N. 1986. An O(EV log V) algorithm for finding a maximal weighted matching in general graphs. SIAM J. Comput. 15, 1, 120–130.
>
> This will be added explicitly in the paper; we completely agree with the reviewer on stating this algorithm explicitly.
>
> - Can the Hungarian algorithm (or any other faster sequential algorithm, or even another parallel algorithm) yield better results here?
>
> Answer:  As the reviewer rightfully mentioned, there are faster algorithms for special cases of MWM, such as on bipartite graphs or unweighted graphs. Such an algorithm can indeed improve the running time bound for the corresponding special case of the problem, as choosing which sequential algorithm to apply can be done independently of the other components of the algorithm.
> Regarding the Hungarian algorithm, we are aware that it works solely for bipartite graphs and indeed would probably improve the bound to O(k^3 polylog(n)) for bipartite graphs only. This bound is also possible for unweighted graphs due to
>
> S. Micali and V. Vazirani, “An O( \sqrt(|V|) · |E|) algorithm for finding maximum matching in general graphs,” in Proc. 21st IEEE Symposium on Foundations of Computer Science (FOCS), 1980, pp. 17–27.
>
>
>
>
>
>
>  - Has the k-MWM problem been proposed and studied outside of your zero communication scenario?
>
> Answer: Yes, the problem and its variants have been studied in various computational models. We agree with the reviewer that this should be added to our paper, and we will do so in the full version of the paper. The following are examples of papers studying matching with small maximum matching size on a standard computational model:
>
> Lahn N, Raghvendra S, Ye J. A faster maximum cardinality matching algorithm with applications in machine learning. Advances in Neural Information Processing Systems. 2021 Dec 6;34:16885-98.
>
> N. Lahn and S. Raghvendra, A weighted approach to the maximum cardinality bipartite matching problem with applications in geometric settings, Journal of Computational Geometry, 11 (2021). Special Issue of Selected Papers from SoCG 2019.
>
> Also, there is abundant research on sparsifying graphs in the context of matching. Specifically, given a graph, a sparsifier is a graph with a smaller number of edges/vertices that preserves the maximum matching size. See the reference below as an example. This topic is mostly relevant when k is relatively small, and is given as additional proof for the necessity of studying this setting of MWM.
>
>  Sepehr Assadi and Aaron Bernstein. Towards a unified theory of sparsification for matching problems. Symposium on Simplicity in Algorithms (SOSA), 2019.
>
>
>
>
>
> - “…As a result, you have to properly motivate that specific problem, which I see no discussion of; you only motivate MWM in a somewhat vague / general sense…”
>
>  Answer: We agree with the reviewer on further motivating the study of k-MWM rather than MWM. While we do agree with the reviewer on adding applications of MWM with small k, we believe the formulation of k-MWM rather than MWM is natural and is in fact a generalization of the problem by allowing the flexibility of choosing a large value of k. Namely,
>
> (k can be arbitrary in the zero-comm model): As explained on Page 3 of the paper, in the zero communication model, the fact that each server receives the cardinality constraint is a more general setting and can be assumed without the loss of generality: We can always set k >= |E|, and the parameter does not affect the algorithm. Yet, all of the results still hold.
>
> (We can avoid sending k in the parallel algorithm): One additional important note that will be highlighted in the paper is that our parallel algorithm works even if the parameter k is not given, so we can tackle arbitrary MWM instances. This can be simply done by running our algorithm with increasing parameters k = 0,1,..., iteratively as long as we return a feasible solution of size sufficiently large (constant factor from k). This augments the running time only by a factor of O(k) and shows that our algorithm works for general MWM instances, rather than only the considered version where a specific parameter is given. We do note that when k = Omega(n), this algorithm is no longer better than the sequential algorithm applied in the last step (any deterministic parallel algorithm can also be used here instead). We will add a more formal version of this discussion to the full version of the paper.
>
>
> - Finally, I don’t find the parallel result for MWM to be particularly surprising, and I don’t think it is at all tight
>
> Answer: Our parallel algorithms for MWM is indeed given as a reduction from a black-box zero-communication algorithm and is not overly involved. The running time for MWM is poly(k)*polylog(n); indeed, the exact poly(k) factor may not be tight, though improving it would likely lead to a faster algorithm for general MWM. This algorithm is foremost interesting because it is completely deterministic, unlike existing polylog(n) parallel algorithms (mentioned by the reviewer). We note that removing the poly(k) factor for a deterministic parallel algorithm would solve a central open question of over four decades.
>
> We will also faithfully change the paper according to the other comments of the reviewer.

---

> > ### Comment · Reviewer_SpuD · 2025-08-01
> >
> > Paper authors,
> >
> > Thank you for your response. I would like to continue discussion related to your parallel bound for MWM. This is a long comment, so I've had to split it into two separate comments due to length limitations.
> >
> > First of all, I must admit that, while reading the paper, my headspace was primarily focused on bipartite matching, although I now realize that oversight upon seeing the mention of the Micali-Vazarani result that general matching is the focus. The results of the paper, of course, work either way, although the distinction is particularly important when discussing the black box invocation at the end of your algorithm. You are right that the Hungarian algorithm doesn't solve the general (non-bipartite) matching problem, so it can't be used directly to solve your final problem of size $k$. However, it doesn't seem like you are unaware of how classical algorithms for min / max weight bipartite matching have been generalized to the min / max general matching problem that you are studying. In particular, in your response you cited this result as being the best for the unweighted problem:
> >
> > S. Micali and V. Vazirani, “An O( \sqrt(|V|) · |E|) algorithm for finding maximum matching in general graphs,” in Proc. 21st IEEE Symposium on Foundations of Computer Science (FOCS), 1980, pp. 17–27.
> >
> > while you cite that the best result for the weighted version is:
> > Galil, Z., Micali, S., and Gabow, H. N. 1986. An O(EV log V) algorithm for finding a maximal weighted matching in general graphs. SIAM J. Comput. 15, 1, 120–130.
> >
> > However, this weighted result was improved just a few years later by Gabow and Tarjan:
> >
> > Gabow, H. N., & Tarjan, R. E. (1991). Faster scaling algorithms for general graph matching problems. Journal of the ACM (JACM), 38(4), 815-853.
> >
> > This paper is a generalization of the 1989 result of Gabow and Tarjan for bipartite graphs, and effectively matches that time bound, roughly $\tilde{O}(m\sqrt{n})$. If nothing else, you should be using a variant of this algorithm for your black box (a simple change) instead of the algorithm you have stated.
> >
> > However, there is more detail here needed. Even if you use the Galil et. al. result you mentioned, it, as described, computes a maximum-weight matching overall (k = n). However, your algorithm, when invoking it as a black box, has only $k^2$ edges, of which at most $k$ must be selected. So, how are you using the Galil et. al. result to compute k-MWM on your instance? As stated, you say you compute "MWM", but is not k-MWM necessary here for this final step of the parallel algorithm? Are you not computing $k$-MWM on a graph of $k^2$ edges?
> >
> > To be clear, I \emph{do} expect the Galil et. al. result can be used to compute $k$-MWM, not just MWM, by simply stopping the algorithm early. It is simply a variant of Edmond's blossom algorithm with an additional data structure to facilitate augmenting path searching on a sparse graph, and the running time matches that of the Hungarian algorithm. The result of Galil et. al. is really a generalization of the Hungarian algorithm for general matching. I know that the Hungarian algorithm computes an optimal cardinality $k$ matching. However, the proof requires some additional details that are not necessarily straightforward at first. Similarly, I suspect the result of Galil et. al. can be applied for $k$-MWM, but again, you need to include those details, which become even more complex due to the tricky nature of general matching vs. bipartite matching.
> >
> > Furthermore, if the result of Galil et. al. is applied to computing an optimal cardinality $k$ matching, then I suspect the running time will become $|E|k\log{|V|}$, resulting in a bound of $\tilde{O}(k^3)$. This is analogous to the use of the Hungarian algorithm in a similar way, as I previously suggested. Effectively, both algorithms aim to find augmenting paths of minimum (or alternatively maximum) "cost". Each augmenting path increases the matching cardinality by 1. Each search takes $O(|E|\log{|V|})$ time. Hence my concern about the tightness of your parallel bound. Indeed, I suspect further improvements to the bound are possible, again, by using the analysis from Gabow and Tarjan.

---

> ### Comment · Reviewer_SpuD · 2025-08-01
> **A continuation of my previous comment regarding parallel matching running times.**
>
> The good news is that these results should lead to improvements to your parallel bound, at least to some extent. However, it does seem as if additional details need to be included, even for the $O(k^4)$ bound as stated. Not all MWM algorithms generalize to $k$-MWM. I believe that the one you cited does, but additional explanation should be included. I am concerned that, if the authors are not familiar enough with the details of this algorithm to give the $\tilde{O}(k^3)$ bound instead of $\tilde{O}(k^4)$, then proving the Galil et. al. algorithm works correctly for $k$-MWM at all might be non-trivial. However, if that is the case, then it is not a result that should be cited (or omitted) as part of the parallel running time bound without additional information.
>
> Finally, it is worth noting that MWM can be computed in time $O(n^\omega W)$ time, where $W$ is the largest edge cost. Unfortunately, I think this algorithm might only work for perfect matching, not cardinality-$k$ matching. However, I feel like it is worth mentioning, so that you are aware of it:
>
> Sankowski, P. (2009). Maximum weight bipartite matching in matrix multiplication time. Theoretical Computer Science, 410(44), 4480-4488.
>
> Authors, please review these additional results and ideas I have mentioned and let me know your thoughts. I understand that this is a lot of discussion regarding the parallel algorithm specifically, and that there is a lot more to the paper than just that one part. However, I think getting the details right for the parallel $k$-MWM is important.  Of particular note, the correctness of your parallel algorithm in some sense requires a $k$-MWM algorithm -- can you either cite an existing published result (paper, theorem, claim, etc.) that generalizes the Galil et. al. result to this problem or give a more comprehensive argument regarding this generalization?
>
> I have read your other comments, and will take them into account as well, but I wanted to begin the discussion of this topic so that you have time to look at the papers I've mentioned.

---

> > ### Author Response · Authors · 2025-08-03
> >
> > We sincerely thank the reviewer for the additional correspondence. We are aware of Gabow and Tarjan's later algorithm. The running time of their algorithm is \tilde{O}(\sqrt(n \cdot \alpha (m,n) \log n)) \cdot m \cdot \log (n \cdot N)), where n,m are the number of vertices and edges, respectively, and N is the largest weight (assume all weights are positive without the loss of generality), and \alpha is the inverse of Ackermann's function. Thus, for exponential weights such as N = 2^n, this algorithm seems inferior to the earlier work of Galil et al. and was not mentioned for this reason (please feel free to correct us if we missed anything here). We also thank the reviewer fo pointing to the paper of Sankowski. However, this result is inferiour for larger weights. We will add this algorithm to a section that will explain the various options for selecting the black-box algorithm raised during our discussion with the reviewer.
> >
> > Regarding obtaining a k-MWM rather than an MWM of size that can be up to k^2. As mentioned by the reviewer, theoretically speaking, it has to be proven that the black-box algorithm can return an optimal k-MWM, rather than a larger MWM.
> > The work of Galil et al. and also the following algorithm of Gabow (with slightly better running time in terms of polylog n factors):
> >
> > Gabow HN. Data structures for weighted matching and nearest common ancestors with linking. In Proceedings of the first annual ACM-SIAM symposium on Discrete algorithms 1990 Jan 1 (pp. 434-443).
> >
> >  are implementations of Edmonds' algorithm. In fact, in the above paper Gabow mentions explicitly (see the second paragraph of the above paper):
> >
> > "A minimum cost matching is a matching of smallest possible cost. There are a number of variations: a minimum cost maximum cardinality matching is a matching with the greatest number of edges possible, which subject to this constraint has the smallest possible cost; minimum cost cardinality k matching (for a given integer k); maximum weight matching; etc. The weighted matching problem refers to all of the problems in this list."
> >
> > Thus, Gabow claims that the result for weighted matching in this paper applies to minimum (analogously, maximum) cost cardinality k matching, which is equivalent to k-MWM in the maximization version. Indeed, this algorithm computes a single search of Edmond's algorithm in time O(m+n log n). We agree with the reviewer that this requires further details, but this seems to indicate that for small k, the running time bound of our black-box algorithm can be improved as suggested by the reviewer. Once we verify the details thoroughly, we will add it to the paper explicitly.
> >
> > P.S., we would like to add the following clarification for a sentence from the rebuttal response:
> > "(k can be arbitrary in the zero-comm model):... Yet, all of the results still hold."
> > This sentence is true only for MW-IS and not for MWM, i.e., each server has to know k in the MWM in the zero communication model. We note that this is written correctly in the paper itself and sincerely apologize for this error.

---

> > > ### Comment · Reviewer_SpuD · 2025-08-03
> > >
> > > Paper authors,
> > >
> > > Thank you for your response, and the clarifications. Regarding the logarithmic dependency on $C$ in the Gabow-Tarjan algorithm, if $C = 2^n$, then at that point, practically speaking, most implementations would require $\log{C}$ bits anyway. As such, a logarithmic dependence on $C$ is implicit to most implementations anyway, if $C$ goes beyond the word size. Nevertheless, I agree that, from a theoretical standpoint, the distinction between strongly and weakly polynomial time bounds should be made. I would recommend simply stating both bounds with regards to your algorithm.
> > >
> > > Thank you for agreeing to check on the improved running time bounds for sequential $k$-MWM. The quote by Gabow and Tarjan, while not necessarily as direct as I would have liked, seems to confirm what I already thought was the case. I trust that there is time for the authors to validate the claim more thoroughly before the camera ready version.
> > >
> > > One more small comment. The authors previously mentioned including the following results as justification for when $k$-MWM was previously studied:
> > > Lahn N, Raghvendra S, Ye J. A faster maximum cardinality matching algorithm with applications in machine learning. Advances in Neural Information Processing Systems. 2021 Dec 6;34:16885-98.
> > >
> > > N. Lahn and S. Raghvendra, A weighted approach to the maximum cardinality bipartite matching problem with applications in geometric settings, Journal of Computational Geometry, 11 (2021). Special Issue of Selected Papers from SoCG 2019.
> > >
> > > In my opinion, these papers are almost completely irrelevant to the context of this paper, and I question why the authors have even offered to include them.
> > >
> > > I am still not wholly convinced of the practical application of very small $k$, but for the most part the authors have addressed my concerns. I will update my rating.

---

### Official Review · Reviewer_urr5 · 2025-07-02

**Clarity:** 3
**Significance:** 3
**Originality:** 3
**Rating:** 5
**Confidence:** 3

**Summary:**

The authors consider a low-round distributed algorithm for matroid independent set and maximum weighted matching. Their model of computation is called "zero-communication": the input is distributed across machines, the machines report a small message to a central server, and the central server can extract a solution from these small messages. Each message is bounded by size k, where k is the size of the solution. This is very similar to 2-round parallel algorithms, though the second step is more restricted in that it requires a single centralized computation.

These problems are highly interesting, and have been studied in related models of computation (e.g., 2-player 1-round communication, message passing, etc.), but not this model exactly. It could be said that this model is a tad niche, but these problems are also highly fundamental that improvements in more standard models are exceedingly rare. Also of note is that these algorithms are perfect algorithms, not approximations.

Their first result is a k-cardinality maximum weight matroid independent set. It is quite simple: find a solution on each machine, send the solution to a central machine, and then the central machine finds the best solution on what it's given. The interesting aspect of this work is the analysis, which carefully examines the properties of independent sets and cycles in matroids to show that this yields a perfect solution. The final algorithm uses O(sqrt(|E|)) deterministic parallel time where |E| is the size of the edge set on a machine by leveraging existing algorithms on their individual machines. The messages passed contain only k elements each.

The next result uses a similar method for maximum weight matching, though it is insufficient to simply solve the problem on each machine and pass them to a central machine. Instead, they extract what are called "strong sets" of size at most 10k^2 + 1 to pass along to the central machine, and the machine can extract the solution from this. Now the messages are slightly larger, but this is also a more difficult problem to solve. The parallel runtime is O(|E|).

They also show how to adapt these to parallel algorithms - both generally for any subset selection problem and specifically for the two studied problems.

**Questions:**

1. Theorem 1.1 says that the running time of each i can be computed in ... time. Do you mean that the runtime IS this value? Seems weird to be measuring the runtime of a computation of the runtime.

2. Is Theorem 1.3 supposed to be for all k in N?

3. In the proof of Claim 3.5, you only need to cite Claim 3.4 the first time. The second time you use it, you get B_i' is independent because you took B_i \cup {e} and removed an element e'' in its only cycle, therefore leaving an independent set. Therefore you're not really using it a second time as implied by how you wrote it. This is correct, right?

4. In the proof of Claim 3.5, once you have B_i' is an independent set, why is it a basis? Is it true that any independent set of a matroid with the same cardinality as a basis must be a basis (due to the second property of matroids), and that means B_i' is a basis?

5. Has this zero-communication model been studied before, or are you introducing it? I think it's well-motivated either way, but I don't know if "zero-communication" is a misleading term term, since there is certainly communication, even if it's only from machines to a central server. It's extremely similar to a 2-round parallel algorithm, but instead you have the second round is a single centralized computation instead of more parallel computation. Do you think there's a more appropriate name for this?

6. Your explanation at the beginning of section 4 is a bit lacking. MWM is not a matroid problem, so you can't necessarily transfer the results from section 3. However, the process you describe does sound like a zero communication algorithm. I'm assuming the issue is, if you distribute the graph across machines and find the k-MWM on each machine, you can't then derive an MWM from the set of solutions on across machines. Is this correct? Or otherwise, why is what you've described not a zero communication algorithm?

7. Can you add some intuition of why the strong set helps you determine the maximum weight matching?

8. Can you describe at a high level what the algorithm for 5.1 looks like?

**Ethical Concerns:**

["NO or VERY MINOR ethics concerns only"]

**Final Justification:**

I have updated the score to be an accept due to the authors' clarifications. I think it is a technically complicated work in an interesting and practical area of study that deserves acceptance.

**Limitations:**

Yes

**Quality:**

4

**Strengths And Weaknesses:**

This paper studies very interesting problems and is generally clear though quite dense at times. The model used is a bit niche (and possibly not studied before), but these are extremely fundamental problems, so progress in more standard models is more difficult. For that reason, I think this work is interesting and important.

My overall rating comes mostly from gaps in my understanding or explanations in the paper itself. If the authors clear these up, I am happy to increase my score.

---

> ### Author Rebuttal · Authors · 2025-07-30
>
> We thank the reviewer for their helpful comments and suggestions. We answer their questions below.
>
>
> 1. Theorem 1.1 says that the running time of each i can be computed in ... time. Do you mean that the runtime IS this value? Seems weird to be measuring the runtime of a computation of the runtime.
>
> Answer: We thank the reviewer for spotting this typo. Indeed, the text should be rewritten as: “...the running time is…”
>
> 2. Is Theorem 1.3 supposed to be for all k in N?
>
> Answer: In the proof of Theorem 1.3 we use: Let any n,m,k,c be natural numbers such that n = c * k * m and m = k. Thus, the theorem holds for any n,m,k as long as m = k and n = c * k * m, i.e., k is equal to the number of servers and n can be arbitrarily large w.r.t. k. We will clarify this exact formulation of Theorem 1.3 in the paper body.
>
>
> 3. In the proof of Claim 3.5, you only need to cite Claim 3.4 the first time. The second time you use it, you get B_i' is independent because you took B_i \cup {e} and removed an element e'' in its only cycle, therefore leaving an independent set. Therefore, you're not really using it a second time as implied by how you wrote it. This is correct, right?
>
> Answer: Yes, this is correct: Since we remove an element from the unique cycle, the remaining elements do not induce more cycles, and we can conclude the next line. We will remove the second reference to Claim 3.4 and add some more details.
>
> 4. In the proof of Claim 3.5, once you have B_i' is an independent set, why is it a basis? Is it true that any independent set of a matroid with the same cardinality as a basis must be a basis (due to the second property of matroids), and that means B_i' is a basis?
>
> Answer: Yes, all bases of a matroid have the same cardinality, and all independent sets with cardinality r (r being the rank of the matroid – the cardinality of a basis) are bases. This indeed stems from the exchange property: Assume that there are two bases X, Y of a matroid such that |X| < |Y|; then, by the exchange property there is y \in Y \setminus X such that X \cup {y} is an independent set in contradiction that X is a basis (an inclusion-wise maximal independent set). Moreover, if one of the independent sets with cardinality r is not a basis, it follows that there exists an independent set with cardinality > r, which means there cannot be any basis of cardinality r using the previous argument.
>
>
> 5. Has this zero-communication model been studied before, or are you introducing it? I think it's well-motivated either way, but I don't know if "zero-communication" is a misleading term, since there is certainly communication, even if it's only from machines to a central server. It's extremely similar to a 2-round parallel algorithm, but instead you have the second round is a single centralized computation instead of more parallel computation. Do you think there's a more appropriate name for this?
>
> Answer: There are many works on bounded or one-directional communication; some of them can be seen as special cases or variants of the zero communication model. However, to the best of our knowledge, the model in its exact form is introduced in this work. We understand the reviewer's concern regarding the model’s name. The reason for choosing this name is that there is no communication between the servers, and they are only able to send messages to the coordinator without receiving any further information. Alternatives we can suggest are:
> - The “in-star” communication model, where the in-star refers to the communication graph with edges only directed from the servers to the coordinator.
> - One-round communication – where we will need to emphasize that the communication is done only from the servers to the coordinator.
>
>
> 6. Your explanation at the beginning of section 4 is a bit lacking. MWM is not a matroid problem, so you can't necessarily transfer the results from section 3. However, the process you describe does sound like a zero-communication algorithm. I'm assuming the issue is, if you distribute the graph across machines and find the k-MWM on each machine, you can't then derive an MWM from the set of solutions on across machines. Is this correct? Or otherwise, why is what you've described not a zero communication algorithm?
>
> Answer: We agree that the first sentences in Section 4 may be slightly misleading, and we will consider removing or clarifying them better. They intended to provide some intuition for why we need a more complicated algorithm for MWM, unlike MW-IS; the analogue of a basis in the context of MWM can be seen as a maximal/maximum weight matching, even though there are crucial differences. Indeed, the suggested algorithm in the first paragraph of Section 4 does describe an algorithm without any communication between the servers, nor backward communication from the coordinator to one of the servers. However, for this to be a successful zero-communication algorithm, the central server must have an optimal solution for the entire instance.
>
> It is quite easy to see why this does not hold using the algorithm selecting a single k-MWM in each server: assume that each server i has two edges (u_i,v_i) and (v_i,w_i); thus, there are two matchings in the input of server i:  The first being the singleton matching {(u_i,v_i)} and the second is {(v_i,w_i)}. Intuitively, server i does not know which of the two matchings would be necessary for the central server. If it sends only one of the matchings, it implies that the central server would not be able to create an MWM.
>
> For example, assume that there are two servers, assume that u_1 = u_2, and assume that all other vertices (i.e., u_1,v_1, w_1, v_2, w_2) are distinct. If Server 1 sends the matching (u_1, v_1) and server 2 sends the matching (v_2,w_2), then the central server cannot construct a matching of size two, even though the union of edges in the two servers contains a size two matching, e.g.,  {(u_1,v_1), (w_2,v_2)}. A zero communication algorithm must be able to obtain an optimal solution in the central server. Thus, sending a single maximal/maximum matching from each server is insufficient for having a zero communication algorithm.
>
>
> 7. Can you add some intuition of why the strong set helps you determine the maximum weight matching?
>
> Answer: We agree with the reviewer that more intuition for strong sets is required. We will add a variation of the following intuition to Section 4.
> Intuition for strong-sets (or, intuition for how to use strong sets to prove Lemma 4.3): In Lemma 4.3, we are given a strong set S_i for each server i and need to prove that the union of S_i’s contains a maximum weight matching (MWM). In the proof of Lemma 4.3, we consider an MWM denoted by M^ that has the maximum size of intersection with the union of S_i’s. If M^ is contained in the union of S_i’s - we are done as the central server can compute M^. Otherwise, there is some element e in M^ and in E_i, for some server i, that does not belong to S_i. Here, we use the fact that S_i is a strong set to find some e^ in S_i such that M’ = M^ \setminus {e} \cup {e^} is an MWM. Specifically, recall the definition of a strong set S_i and the edge e, which guarantees one of the following (Definition 4.1):
> - e is in S_i - we already know this is not the case by the definition of e.
> - There is a matching M contained in S_i of (a sufficiently large) cardinality 2k+1 such that every edge in M has higher or equal weight to the weight of e. Thus, as M is sufficiently large and since M and M^ are matchings, there is e^ in M such that M’ = M^ \cup {e^} is a matching (more specifically, since M^ and M are matchings, each endpoint of M^ can appear at most once as an endpoint of M, and since |M| > 2 |M^| there is an edge in M that can be added to M^ resulting in a matching).  Finally, the weight of M’ can only be higher or equal to the weight of M^ since e^ in M.
> - There is an endpoint x of e and there are 2k+1 distinct edges adjacent to x: (x, v1), . . . ,(x, v_{2 k+1}), each of weight at least the weight of e. Then, in this case, we can use one of these edges as e^ to define M’ = M^* \setminus {e} \cup {e^}  analogous to the above case. In more detail, since |M^| \leq k it has at most 2k endpoints. Therefore, as there are 2k+1 edges as mentioned above, one of these edges e^=(x,v_j) satisfies that v_j is not an endpoint of M^; thus, if we remove e from M^ and add e^, it would induce a matching M’ = M^ \setminus {e} \cup {e^} (as the endpoint x replaces its matched endpoint to a new endpoint not in M^), that will have higher or equal weight to the weight of M^.
> The above two cases yield a contradiction that M^ is an MWM of maximum intersection with the union of S_i’s.
>
> 8. Can you describe at a high level what the algorithm for 5.1 looks like?
>
> Answer: At a high level, the algorithm for 5.1 can be described as follows (see page 26, line 1067 of the full submission including the appendix for the pseudocode):
> - While the number of given elements is larger than 4*f(k):
> - We partition the elements (in parallel) evenly and arbitrarily into roughly |E^j| / 4*f(k) servers, where $E^j$ is the current set of elements (initialized as E, the given set of elements).
> - Execute the black-box zero-communication algorithm A on the instance induced from the above partition (using the initial given parameter k).
> - Update E^{j+1}, the elements for the next iteration, as the union of elements brought by Algorithm A in the above step to the central server.
>
> The intuition is that at each iteration, each server applies filtering of the elements it receives: Each server gets roughly  4*f(k) elements and returns only f(k) elements (in parallel to the other servers). Thus, in O(log n) such rounds, we obtain an O(f(k)) size instance, which can then efficiently be solved using some black-box algorithm.

---

> > ### Comment · Reviewer_urr5 · 2025-08-08
> >
> > This is all very helpful, thanks for the clarifications. I will update my score.

---

### Official Review · Reviewer_wKn3 · 2025-07-03

**Clarity:** 4
**Significance:** 3
**Originality:** 2
**Rating:** 4
**Confidence:** 4

**Summary:**

This paper considers the cardinality-constrained version of the maximum-weight independent set problem of matroids and the maximum-weight graph matching problem under the "zero communication" model. In this model, the elements and their weights of the ground set (or the edge set for the matching problem) are partitioned and stored over a multiple number of servers. Every element is in exactly one server. Each server processes its data, and sends a small message to the coordinator, which must use these messages to produce the final output.

For the maximum-weight independent set problem, the answer is intuitive: each server finds the solution for the matroid restricted to the elements the server has, and sends this solution to the coordinator. Since the problem is cardinality-constrained, each server will send at most $k$ items to the coordinator. The coordinator then finds the solution for the matroid restricted to, this time, the union of the solutions received from the servers. Its correctness is shown using an exchange argument on bases.

For the maximum-weight matching problem, the message is larger than before: each server sends $O(k^2)$ edges to the coordinator. The intuition is that a server needs to send to the coordinator only those edges that are valuable enough to be one of the k edges chosen in the final solution. As such, an edge is "useful" only if every cardinality-$(2k+1)$ matching contains the edge and it is one of the $(2k+1)$ most valuable edges incident with each endpoint. The number of these useful edges is $O(k^2)$, as can be easily seen from an argument that a high-degree graph with many edges must have a large matching.

Based on these two zero-communication results, the paper also gives a parallel algorithm. The algorithm partitions the ground/edge set into small pieces to achieve $\mathrm{poly}(k)\cdot \log(n)$ running time. The paper also presents lower bound results that complement the main results. Finally in the appendix, it also shows that the shortest path problem cannot be solved without sending the entire set of edges to the coordinator.

**Questions:**

- Although the paper states on Page 3 that the version with the cardinality constraint is more general, using the reduction may blow up the size of the message. It seems that messages with size $O(r)$ is sufficient for the independent set problem, where $r$ denotes the rank of the matroid, but what can we do for the matching problem?

- The model assumes that the matroid structure is known by all servers (at least when restricted to its elements) and the coordinator. How practical is this assumption?

Let me write a few minor comments here too:

- The abstract does not make it clear that k is given as input.

- Line 55, (probability 1): even if an algorithm succeeds with probability 1, it may still be randomized.

- "pseudo-polynomial" sounds somewhat misleading. How about saying "polylogarithmic in the size of the ground set"?

**Ethical Concerns:**

["NO or VERY MINOR ethics concerns only"]

**Final Justification:**

Most of my questions have been satisfactorily answered. Authors also discussed weaknesses I raised, and including these discussions in the paper would greatly improve it.

However, for Question 1, we don't seem to have a good answer, according to our discussion during the rebuttal period. I believe that requiring the cardinality constraint as part of input is somewhat artificial, especially considering that it turned out to be not easy to eliminate the assumption for MWM. As such, I find that my concern regarding Weakness 1 hasn't been resolved, and I'd maintain my original rating.

**Limitations:**

Yes. While the paper clearly states assumptions it makes for its theoretical results, I discussed some additional concerns above (see Questions).

**Paper Formatting Concerns:**

None.

**Quality:**

3

**Strengths And Weaknesses:**

### Strength

- The zero communication model is an interesting model and the paper proposes to consider fundamental problems of combinatorial optimization under this model.

- Presentation is simple and clear.

### Weakness

- (See questions.) The algorithm requires a cardinality constraint to keep the messages small. This seems somewhat artificial.

- As was shown by the paper itself, the model is too restrictive to solve even simple problems like the shortest path problem. It would be nice to see some more arguments why the zero communication model should be studied (e.g. as a means to obtain good parallel algorithms).

- A suggestion: the analysis of the two algorithms, especially the one for the independent set problem, uses techniques that are quite standard. Can we omit some of these arguments from the first nine pages and instead bring some of the matrials in the appendix?

---

> ### Author Rebuttal · Authors · 2025-07-30
>
> We thank the reviewer for their helpful comments and suggestions. We answer their questions below.
>
> - The algorithm requires a cardinality constraint to keep the messages small. This seems somewhat artificial.
>
> Answer: We distinguish between two settings: (1) where k is provided, and (2) where it is not. Even though our paper is formalized according to (1), we explain below the implications for case (2).
>
> (k can be arbitrary in the zero-comm model): As explained on Page 3 of the paper, in the zero communication model, the fact that each server receives the cardinality constraint is a more general setting and can be assumed without the loss of generality:  We can always set k >= |E|, and the parameter does not affect the algorithm. Yet, all of the results still hold; for example, we would still get an r-data algorithm for MW-IS where r is the rank of the matroid. For MWM with no parameter given, our results are Theta(OPT^2)-data algorithm and a matching lower bound, where OPT is the size of a maximum matching, which gives a tight bound up to a constant. If OPT is sufficiently small, this result shows we can avoid sending all edges to the coordinator. However, if OPT = Omega(n), then we may be forced to send the entire set of edges (in this case, the trivial algorithm of sending all edges matches the lower bound).
>
> (We can avoid sending k in the parallel algorithm): One additional important note that will be highlighted in the paper is that our parallel algorithm works even if the parameter k is not given, so we can tackle arbitrary MWM instances. This can be simply done by running our algorithm with increasing parameters k = 0,1,..., iteratively as long as we return a feasible solution of size sufficiently large (constant factor from k). This augments the running time only by a factor of O(k) and shows that our algorithm works for general MWM instances, rather than only the considered version where a specific parameter is given. We do note that when k = Omega(n), this algorithm is no longer better than the sequential algorithm applied in the last step (any deterministic parallel algorithm can also be used here instead). We will add a more formal version of this discussion to the full version of the paper.
>
> (k-MWM is a generalization of MWM): The parameter k allows us to tackle the more general problem with the cardinality constraint, which we find interesting, i.e., to apply the algorithm on matroid/graphs with a large optimum size, but we want to restrict ourselves only to a smaller-sized solution. There is a practical motivation for that in, for example,  unbalanced bipartite graphs, which can represent in practice scheduling (many) tasks to k processors, or allocating ads to (a few) advertisement slots, and many more practical examples of unbalanced assignment problems.
>
>
> - As was shown by the paper itself, the model is too restrictive to solve even simple problems like the shortest path problem. It would be nice to see some more arguments why the zero communication model should be studied (e.g., as a mean to obtain good parallel algorithms).
>
> Answer: We believe the zero-communication model is worth studying for several key reasons. First, indeed, as mentioned by the reviewer, we showed its close connection to parallel algorithms. Second, it allows a clean setting to distinguish between the difficulty of problems in a common distributed setting.  We can think of problems that fit well in this model as “communication-efficient” – that can be distributed with small/no communication overhead. Knowing which problems exhibit this property can have practical implications and enable robust algorithms that do not have to process large data on a single processor.
> Thus, “easy” problems like shortest paths in a classic computational model that are hard in this model show the usefulness of the model in distinguishing problems according to a restrictive model of communication complexity.
> Finally, the model generalizes well-studied problems in the literature, such as one-direction communication, making it a natural extension of prior work (as discussed in the introduction).
>
>
>
>
>
> - A suggestion: the analysis of the two algorithms, especially the one for the independent set problem, uses techniques that are quite standard. Can we omit some of these arguments from the first nine pages and instead bring some of the matrials in the appendix?
>
> Answer: We thank the reviewer for this suggestion, and these parts can be moved to the appendix. We will make this change.
>
> - The model assumes that the matroid structure is known by all servers (at least when restricted to its elements) and the coordinator. How practical is this assumption?
>
> Answer: For zero communication algorithms, it suffices that each server can compute independence only on the subsets of elements that the server received. Without this assumption, it would be impossible to design a zero-communication algorithm that does not send all elements.
> Most matroids considered in the literature and those with the strongest applications are linear matroids, which can be compactly defined and allow for computing independence, and representing a subset of elements in such matroids requires relatively small memory.
> For example, consider the important family of graphic matroids, where the elements are edges of a graph and independent sets are acyclic subsets of edges. By taking a subset S of edges to a server i, we only need the server to know the contraction of the given graph to the set of edges S, which is of the same order as the number of edges in S (e.g., if the edges given to a server are only {1,2}, {2,3}, {1,3},  then even if there are n>>3 vertices and m>>3 edges overall in the graph, we can represent this matroid by 3 vertices and 3 edges and verify independence using a cycle detection algorithm on this subgraph only).
>  However, there are also non-linear matroids (e.g., the Vamos matroid); a non-linear matroid may require exponential memory (in the number of elements of the ground set) to distinguish correctly between independent sets and non-independent sets. These matroids appear less in practical settings and will require that each server have larger memory.
> The abstract does not make it clear that k is given as input.
> We guarantee to add that to the abstract.
>
>
> - Line 55, (probability 1): even if an algorithm succeeds with probability 1, it may still be randomized.
>
> Answer: We thank the reviewer for this distinction. This will be rewritten.
>
>
> - "pseudo-polynomial" sounds somewhat misleading. How about saying "polylogarithmic in the size of the ground set"?
>
> Answer: Thank you for the suggestion. We will make this change as well.

---

> > ### Comment · Reviewer_wKn3 · 2025-08-02
> >
> > Thank your for the response. I still have a question about what happens when $k$ is not known in advance.
> >
> > For MW-IS, it is easy for many reasons. Given a server, if we pretend that we run the algorithm once for $k_1$ and then once for $k_2$, for $k_1>k_2$, the set of elements chosen by the first algorithm can be assumed to be a superset of the latter. Alternatively, we can just use the fact that, globally, an optimal solution for $k_2$ is contained in every optimal solution for $k_1$.
> >
> > For MWM, there doesn't seem to be an easy "blackbox" arguments. A strong set for a larger $k$ is not necessarily a superset of a strong set for a smaller $k$, and a solution for a larger $k$ is not necessarily a superset of a solution for a smaller $k$. The algorithm in lines 331-337 also does not have such monotonicity. I can see how one could get $O(\\min(|V|,|E|)^3)$-data algorithm (basically by sending all data for every possible $k$), but anything better doesn't seem too trivial. (Even if we can show that a size-$\\Theta(\\mathsf{OPT}^2)$ information exists that can be sent to the coordinator, another issue is that the servers don't know what $\\mathsf{OPT}$ is, so that data needs to be computable without that knowledge.) Can you explain why "[for] MWM with no parameter given, [your] results are $\\Theta(\\mathsf{OPT}^2)$-data algorithm"?

---

> > > ### Author Response · Authors · 2025-08-02
> > >
> > > Thank you for the clarification, you are right. Our claim in the rebuttal response, mentioning that the parameter can be removed for MWM, is incorrect. The claim for MW-IS does hold, as indicated by the reviewer. Indeed, a single server whose sub-graph is a star cannot know which subset of edges to send: if it does not send a sufficient number of edges, all of these edges might belong to the optimal solution and cannot be used, unlike some other edge from the star. Note that this claim only appears in the rebuttal response and is not part of the submitted paper. We will add to the paper this distinction between MW-IS and MWM.
> > >
> > > We sincerely apologize for this mistake and thank the reviewer again for spotting this.

---

> > > > ### Comment · Reviewer_wKn3 · 2025-08-06
> > > >
> > > > Thank you for the comments. I'm inclined to maintain my score, since I still feel that requiring the cardinality constraint as part of input is somewhat artificial, especially considering that it is not easy to eliminate the assumption for MWM.

---

> > > > > ### Author Response · Authors · 2025-08-07
> > > > >
> > > > > We thank the reviewer for the discussion. The reviewer's comments will be taken into account in the final version of the paper.

---

### Note · Authors · 2025-08-15

We sincerely thank the reviewers for thoroughly reading the paper, for all helpful comments and suggestions, and the rebuttal discussion. In the final version of the paper, we will incorporate all suggestions and comments raised in the reviews and during the rebuttal discussion. Key improvements will include:

(1) More intuition about the constructions in the matching algorithm.

(2) A subsection explaining in detail the various explicit options for selecting the sequential algorithm used in the parallel algorithm (Section 5), including faster algorithms for special cases such as bipartite graphs.

(3) Stronger motivation with real-world examples for maximum matching instances with small matching cardinality (such as unbalanced bipartite graphs, arising in scheduling, marketing, ad allocation, etc.).

---

### Decision · Program_Chairs · 2025-09-17

**Decision:**

Accept (poster)

**Comment:**

All reviewers gave positive scores; some appreciated the paper's theoretical contribution and writing quality. The authors successfully addressed several concerns, leading to higher scores. While there were initial doubts about the paper's fit for NeurIPS, the discussion concluded that this should not be a reason for rejection.